# Investigation of *Drosophila fruitless* neurons that express Dpr/DIP cell adhesion molecules

**Savannah G Brovero[1][†], Julia C Fortier[1][†], Hongru Hu[1][†], Pamela C Lovejoy[1][†][‡], Nicole R Newell[1][†], Colleen M Palmateer[1][†], Ruei-Ying Tzeng[1][†], Pei-Tseng Lee[2], Kai Zinn[3], Michelle N Arbeitman[1]***

[1]Department of Biomedical Sciences and Program of Neuroscience, Florida State University, College of Medicine, Tallahassee, United States; [2]Department of Molecular and Human Genetics, Baylor College of Medicine, Houston, United States; [3]Division of Biology and Biological Engineering, California Institute of Technology, Pasadena, United States

**\*For correspondence:**
michelle.arbeitman@med.fsu.edu

[†]These authors contributed equally to this work

**Present address:** [‡] Department of Biology, St. Joseph's College, New York, United States

**Competing interests:** The authors declare that no competing interests exist.

**Abstract** *Drosophila* reproductive behaviors are directed by *fruitless* neurons. A reanalysis of genomic studies shows that genes encoding *dpr* and *DIP* immunoglobulin superfamily (IgSF) members are expressed in *fru P1* neurons. We find that each *fru P1* and *dpr/DIP* (*fru P1* ∩ *dpr/DIP*) overlapping expression pattern is similar in both sexes, but there are dimorphisms in neuronal morphology and cell number. Behavioral studies of *fru P1* ∩ *dpr/DIP* perturbation genotypes indicate that the mushroom body functions together with the lateral protocerebral complex to direct courtship behavior. A single-cell RNA-seq analysis of *fru P1* neurons shows that many *DIPs* have high expression in a small set of neurons, whereas the *dprs* are often expressed in a larger set of neurons at intermediate levels, with a myriad of *dpr/DIP* expression combinations. Functionally, we find that perturbations of sex hierarchy genes and of *DIP-ε* change the sex-specific morphologies of *fru P1* ∩ *DIP-α* neurons.

## Introduction

A goal of neuroscience research is to gain molecular, physiological and circuit-level understanding of complex behavior. *Drosophila melanogaster* reproductive behaviors are a powerful and tractable model, given our knowledge of the molecular-genetic and neural anatomical basis of these behaviors in both sexes. Small subsets of neurons have been identified as critical for all aspects of reproductive behaviors. These neurons express sex-specific transcription factors encoded by *doublesex* (*dsx*) and *fruitless* (*fru*; *fru P1* transcripts are spliced under sex hierarchy regulation; *Figure 1A*) (reviewed in *Dauwalder, 2011*; *Yamamoto et al., 2014*; *Andrew et al., 2019*; *Leitner and Ben-Shahar, 2020*). *dsx*- and *fru P1*-expressing neurons are present in males and females in similar positions, and arise through a shared developmental trajectory (*Ren et al., 2016*), although these neurons direct very different behaviors in males and females. Males display an elaborate courtship ritual that includes chasing the female, tapping her with his leg, and production of song with wing vibration (reviewed in *Greenspan and Ferveur, 2000*). The female decides whether she will mate and then, if mated, displays post-mating behaviors that include egg laying, changes in diet, and changes in receptivity to courtship (see *Laturney and Billeter, 2014*; *Aranha and Vasconcelos, 2018*; *Newell et al., 2020*).

Sex differences in the nervous system that contribute to reproductive behaviors include dimorphism in *dsx* and *fru P1* neuron number, connectivity, and physiology. The molecules and mechanisms that direct these differences are beginning to be elucidated. Several genome-wide studies

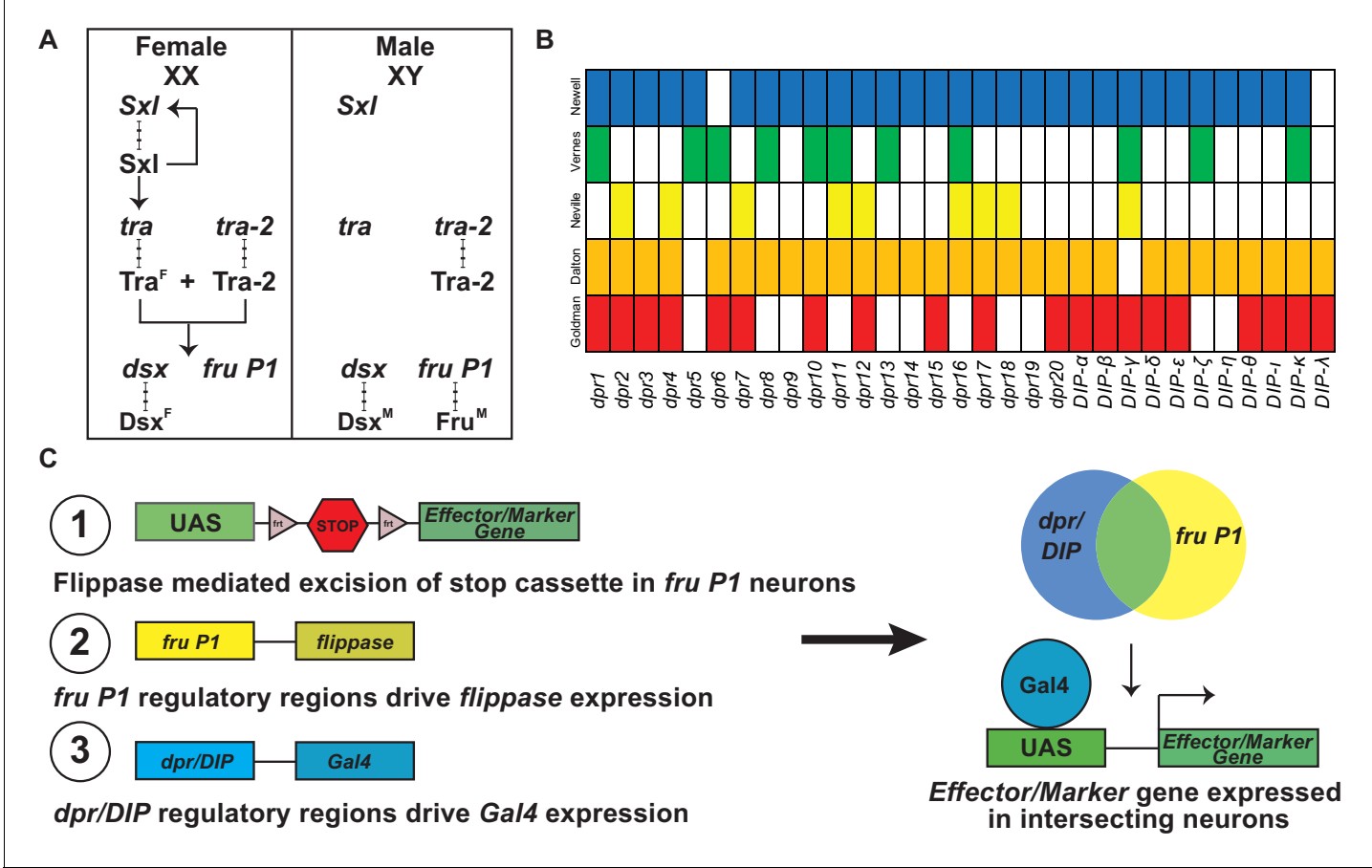

**Figure 1.** Overview of sex hierarchy and experimental design. (**A**) The *Drosophila* somatic sex determination hierarchy is an alternative pre-mRNA splicing cascade (reviewed in *Andrew et al., 2019*). The presence of two X chromosomes in females results in splicing of *Sxl* pre-mRNA, such that functional Sxl is produced. Sxl regulates *Sxl* and *tra* pre-mRNA splicing, resulting in continued production of functional Sxl and Tra in females. Tra and Tra-2 regulate the pre-mRNA splicing of *dsx* and *fru P1* in females, whereas in males *dsx* and *fru P1* are spliced by the default pre-mRNA splicing pathway. The sex-specific splicing results in production of sex-specific Dsx and Fru transcription factors (*Burtis and Baker, 1989*; *Ito et al., 1996*; *Ryner et al., 1996*). *dsx* regulates sex differences that lead to both dimorphic behavior and gross anatomical morphological differences, whereas *fru P1* regulates sex differences that lead to dimorphic behaviors. (**B**) Previous genome-wide studies found that *dpr/DIPs* are regulated downstream of *fru P1*, Fru$^M$, and/or are expressed in *fru P1* neurons (*Goldman and Arbeitman, 2007*; *Dalton et al., 2013*; *Neville et al., 2014*; *Vernes, 2015*; *Newell et al., 2016*). (**C**) A genetic intersectional strategy was used to express marker or effector genes in *fru P1* ∩ *dpr/DIP* neurons. This strategy takes advantage of the two-component Gal4/UAS expression system, and Flippase-mediated removal of a stop cassette within an expression vector. Expression of the marker/effector gene requires both removal of the stop cassette via *fru P1-flippase* (*flp*) expression and expression of Gal4 via *dpr/DIP* regulation. Therefore, only neurons that express both *fru P1* and one of the *dpr/DIPs* have expression of the effector or marker (shown on right).
The online version of this article includes the following source data for figure 1:

**Source data 1.** Data table of Fru$^M$ binding sites in *dpr* and *DIP* genes for three Fru$^M$ isoforms.

have been performed to find genes that are regulated by male-specific Fru (Fru$^M$) or are expressed in *fru P1* neurons. These independent studies examined *fru P1* loss-of-function and gain-of-function gene expression changes, *fru P1* cell-type-specific gene expression, and the direct targets for Fru$^M$ binding (*Goldman and Arbeitman, 2007*; *Dalton et al., 2013*; *Neville et al., 2014*; *Vernes, 2015*; *Newell et al., 2016*). A reanalysis of these genomic studies demonstrates that cell adhesion molecules that are members of the immunoglobulin superfamily (IgSF) are regulated by male-specific Fru (Fru$^M$) or are expressed in *fru P1* neurons (see *Figure 1B*). In this study, we focus on an interacting set of IgSF molecules encoded by *dprs/DIPs*.

The Dpr and DIP proteins are membrane-linked cell adhesion molecules with N-terminal extracellular Ig domains and C-terminal glycosyl-phosphatidylinositol (GPI) linkage sequences or transmembrane domains. The Dpr proteins have two extracellular Ig domains, whereas DIPs have three

(reviewed in *Zinn and Özkan, 2017*; *Sanes and Zipursky, 2020*). In addition to the genomic studies, our previous work showed that *dpr1*, the founding member of the *dpr* family (*Nakamura et al., 2002*), has a role in gating the timing of the male courtship steps (*Goldman and Arbeitman, 2007*). The finding that cell adhesion molecules are regulated by Fru^M fits well with studies that showed that there are sex-differences in arborization volumes throughout the central nervous system (*Cachero et al., 2010*; *Yu et al., 2010*). Thus, it was predicted that differences in neuronal connectivity are one mechanism to generate behavioral dimorphism (*Cachero et al., 2010*; *Yu et al., 2010*).

In-depth, in vitro analyses of Dpr/DIP protein-protein interactions have shown that most DIPs interact with multiple Dprs, and vice versa. Additionally, 4 of the 11 DIPs and 2 of the 21 Dprs display homophilic interactions, and there are two heterophilic interactions between Dprs (*Özkan et al., 2013*; *Carrillo et al., 2015*; *Cosmanescu et al., 2018*). Functional analyses of the Dprs and DIPs have revealed roles in synaptic connectivity and specificity of neuronal targeting in the *Drosophila* neuromuscular junction, visual system, and olfactory system (*Carrillo et al., 2015*; *Tan et al., 2015*; *Barish et al., 2018*; *Xu et al., 2018*; *Ashley et al., 2019*; *Courgeon and Desplan, 2019*; *Menon et al., 2019*; *Venkatasubramanian et al., 2019*; *Xu et al., 2019*). Additionally, cell adhesion molecules have already been shown to be important for sculpting dimorphism in *fru P1* neurons, with studies of the IgSF member encoded by *roundabout* (*robo*) shown to be a direct target of Fru^M and responsible for dimorphic projections and morphology (*Mellert et al., 2010*; *Ito et al., 2016*). Thus, the Dprs/DIPs are good candidates for directing sexual dimorphisms in connectivity and morphology that may underlie differences in reproductive behavior.

Here, we further examine the expression repertoires of *dprs/DIPs* in *fru P1* neurons using immunofluorescence analyses. We examine the sets of neurons that express *fru P1* and one of the *dprs* or *DIPs*, using a genetic intersectional strategy (*fru P1* ∩ *dpr/DIP*; *Figure 1C*). Additionally, this genetic strategy provides a method to examine the roles of neurons expressing *fru P1* and a *dpr* or *DIP* in directing male reproductive behaviors. Here, we elucidate which combinations of neurons lead to atypical courtship behaviors when activated or silenced. A single-cell RNA-seq analysis shows the myriad and unique combinations of *dprs/DIPs* expressed in individual *fru P1* neurons, with expression of at least one *dpr* or *DIP* in every *fru P1* neuron examined. Additionally, the single-cell RNA-seq analyses show that many *dprs* are expressed in a large number of neurons at intermediate levels, whereas most *DIPs* have higher expression in fewer neurons. Genetic perturbation screens reveal functional roles of the sex hierarchy, and *DIP-ε*, in establishing sex-specific architecture of *fru P1* ∩ *DIP-α* neurons.

## Results

### *dprs* and *DIPs* are expressed in *fru P1* neurons, with expression regulated by Fru^M

A reanalysis of previous genomic studies shows that *dprs* and *DIPs* are regulated by Fru^M and are expressed in *fru P1* neurons (*Figure 1B*). These independent studies examined *fru P1* loss-of-function (*Goldman and Arbeitman, 2007*) and Fru^M gain-of-function/overexpression (*Dalton et al., 2013*) gene expression changes, *fru P1* cell-type-specific gene expression in males and females (*Newell et al., 2016*), and identified direct genomic targets of Fru^M (*Neville et al., 2014*; *Vernes, 2015*). The majority of the *dpr/DIP* genes are identified as regulated by Fru^M or expressed in *fru P1* neurons in at least three of these independent genome-wide studies (*Figure 1B*). Furthermore, a computational DNA-binding site analysis confirms Fru^M regulation. There is alternative splicing at the 3' end of *fru P1* transcripts that results in one DNA-binding-domain-encoding-exon being retained out of five potential exons. The predominant isoforms of Fru^M contain either the A, B, or C DNA-binding domain that each bind a different DNA sequence motif (genome-wide analysis described in *Dalton et al., 2013*). When we search for the presence of the three DNA sequence motifs near/in the *dpr/DIP* loci, Fru^M binding sites were found near/in all but two *dpr/DIP* loci that are examined (*Figure 1—source data 1*). Further evidence that *dprs/DIPs* have a role in *fru P1* neurons comes from a live-tissue staining approach, using epitope-tagged, extracellular regions of a Dpr or DIP to examine binding (as done in *Fox and Zinn, 2005*; *Lee et al., 2009*; *Özkan et al., 2013*). This revealed sexual dimorphism in binding of tagged Dpr/DIP proteins to *fru P1* neurons in the subesophageal ganglion brain region, with more neurons with overlap detected in males

(*Brovero et al., 2020*). Together, these results demonstrate that every *dpr* and *DIP* gene is either regulated by Fru$^M$ or expressed in *fru P1* neurons in either males and females.

## A genetic intersectional approach identifies neurons that express both *fru P1* and a *dpr* or *DIP* in males and females

The above results led us to examine the expression patterns in the central nervous system of neurons that express both *fru P1* and a *dpr* or *DIP*, using a genetic intersectional approach (*Figure 1C*). This approach restricts expression of a membrane-bound-GFP marker to neurons with intersecting expression of *fru P1* and a *dpr* or *DIP* (*fru P1* ∩ *dpr/DIP*). This is accomplished using a UAS-membrane-bound GFP reporter transgene that requires removal of an FRT-flanked stop cassette for expression. Removal of the stop cassette is mediated by *fru P1*-driven FLP recombinase (*Yu et al., 2010*). This system is used in combination with a collection of *dpr*- and *DIP-Gal4* transgenic strains (*Figure 1C*; *Venken et al., 2011*; *Nagarkar-Jaiswal et al., 2015a*; *Nagarkar-Jaiswal et al., 2015b*; *Tan et al., 2015*; *Lee et al., 2018*). We primarily focused the analysis on 4- to 7-day-old adults (*Figure 2* and *Figure 3*), which are sexually mature, and 0- to 24-hr adults to determine if the expression patterns change during adult stages (*Figure 2—figure supplement 1* and *Figure 3—figure supplement 1*). Additionally, behavioral studies were performed on 4–7 day adults (*Figure 4*, *Figure 5*, *Figure 6*), so the expression and behavioral data can be co-analyzed (*Figure 7*). At a gross morphological level, the patterns we observe in older 4- to 7-day-old adults are also present in 0–24 hr adults, though in some cases expression in the mushroom body was not robust at the earlier 0–24 hr time point.

Based on our examination of the expression patterns in 27 intersecting genotypes, we find that 24 genotypes showed clear and consistent, membrane-bound GFP expression in the central nervous system. Of these, only two *fru P1* ∩ *DIP* genotypes have restricted and unique patterns (*fru P1* ∩ *DIP-δ* and *fru P1* ∩ *DIP-α*), whereas the other genotypes have broader expression, with many in similar regions/patterns (*Figures 2* and *3*). *fru P1* ∩ *DIP-δ* neuronal projections are near the anterior surface of the protocerebrum and appear to be near the γ -lobe of the mushroom body, based on visual inspection (*Aso et al., 2014*). The *fru P1* ∩ *DIP-α* expression pattern is described in detail below. The 22 intersecting genotypes with broad expression, in both males and females, have consistent expression in the brain lateral protocerebral complex, including within the arch, ring, junction, and crescent (for summary see *Figure 7* and *Source data 1*). This region has been shown to have *fru P1* neurons with sexually dimorphic arbor volumes (*Cachero et al., 2010*; *Yu et al., 2010*). Furthermore, the lateral protocerebral complex has inputs from sensory neurons and is predicted to be a site of sensory integration, to direct motor output necessary for coordinating courtship activity (*Yu et al., 2010*). We find eight intersecting genotypes have expression in mushroom bodies in both males and females. This region has a well-established role in learning and memory, including learning in the context of courtship rejection (*Jones et al., 2018*; *McBride et al., 1999*; *Montague and Baker, 2016*; *Zhao et al., 2018*). Overall, the majority of *fru P1* ∩ *dpr/DIP* genotypes have expression in similar regions, suggesting that some may function in a combinatorial manner within a neuron to direct patterning and/or synaptic targeting, consistent with the single-cell RNA-sequencing data presented below.

We observe sex differences in morphological features and cell body number in regions we scored (*Figures 2* and *3*). These regions were largely chosen because they were previously reported to have *fru P1* neurons that display sex differences downstream of sex hierarchy genes *transformer* and *fru P1* (*Cachero et al., 2010*; *Yu et al., 2010*). For example, 18 intersecting genotypes show consistent presence of signal in the mesothoracic triangle neuronal projections in males, but only two genotypes do so in females. While both males and female have expression in the DA1 and VA1v antennal lobe glomeruli in several intersecting genotypes, we also observe sexual dimorphism, with four genotypes having consistent expression in only female DA1 glomeruli (*fru P1* ∩ *dpr3, dpr10, dpr17, DIP-θ*). In the ventral nerve cord, neurons that cross the midline are consistently observed only in males and not females. Previous work found a midline crossing phenotype that was also male-specific for a set of gustatory neurons (*Mellert et al., 2010*). For all regions where cell bodies are counted, the trend was that there are more cell bodies in males than females. Thus, the differences in the patterns of expression between males and females are not large, with several genotypes having quantitative differences in the numbers of cell bodies present. It is possible that there are additional quantitative differences that are not detected based on the resolution of the analyses,

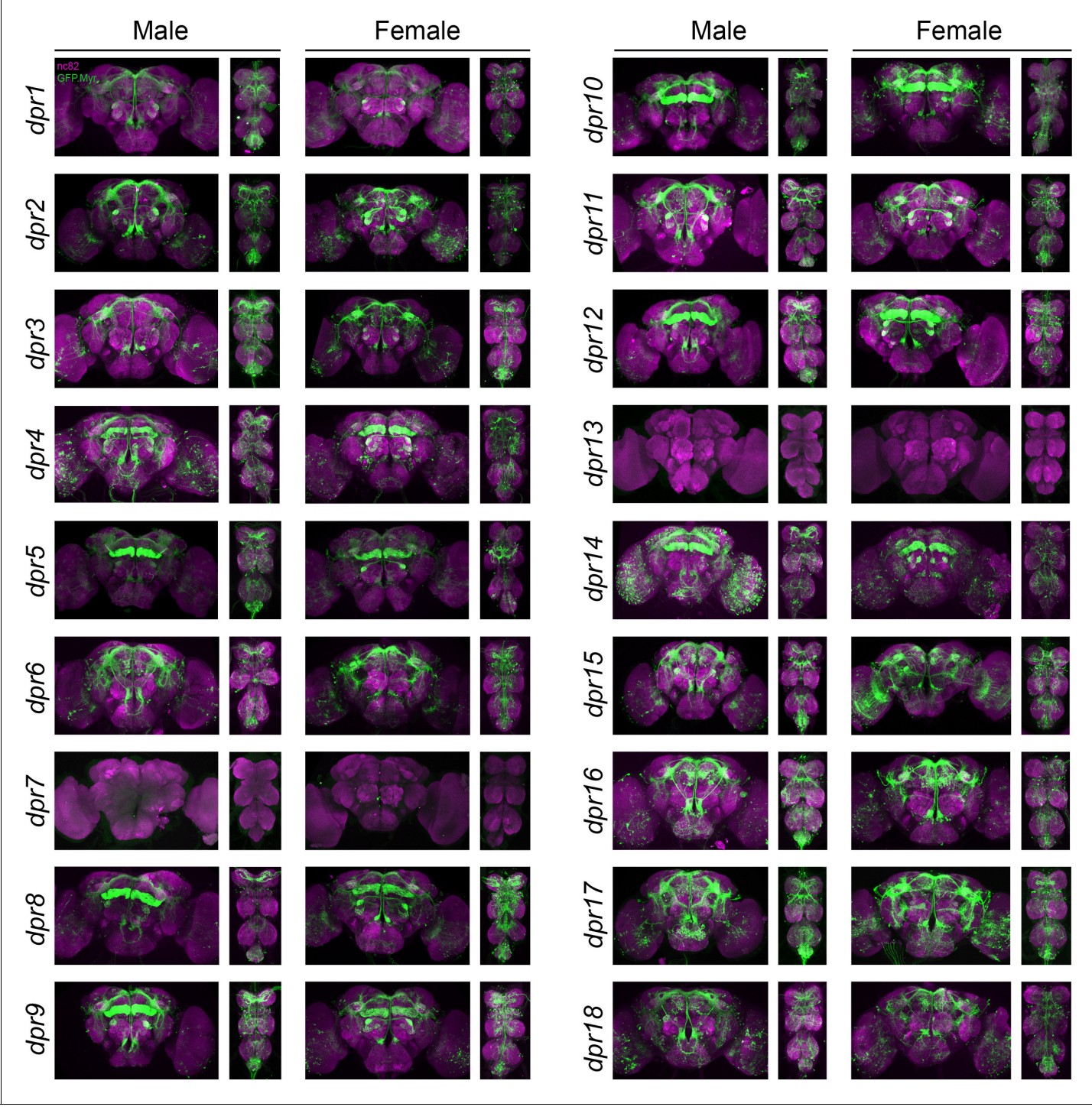

**Figure 2.** Visualization of *fru P1* ∩ *dpr* neurons. Maximum intensity projections of brain and ventral nerve cord tissues from 4- to 7-day-old male and female flies. The *fru P1* ∩ *dpr* intersecting neurons are labeled with green (rabbit α-GFP Alexa Flour 488), and neuropil are labeled with magenta (mouse α-nc82, Alexa Flour 633). The genotype is *dpr-Gal4/10xUAS > stop > GFP.Myr; fru P1$^{FLP}$*, except for *dpr4, dpr14, dpr18, and dpr19*. These *Gal4* transgenic strains were generated using a CRISPR-mediated insertion of the *T2A-Gal4* with the dominant 3xP3-GFP marker. For this strain, *10xUAS > stop > myr::smGdP-cMyc* was used and *fru P1* ∩ *dpr* intersecting neurons are labeled with red (rabbit α-Myc, Alexa Flour 568) and then false-colored to green. The neuropil are labeled with magenta (mouse α-nc82, Alexa Flour 633). These Gal4s did not show consistent *fru P1* intersecting expression: *dpr7, dpr13*, and *dpr19*. The *dpr7* and *dpr13* Gal4s have expression with 10xUASmCD8gfp, confirming the Gal4s can drive expression outside of *fru P1* neurons. *dpr19* was tested with 10xUAS-RFP and did not show expression outside of *fru P1* neurons.

The online version of this article includes the following figure supplement(s) for figure 2:

*Figure 2 continued on next page*

*Figure 2 continued*

**Figure supplement 1.** Visualization of *fru P1* ⋂ *dpr* neurons from 0-to 24-hour adults.

including quantitative differences in expression level of *dpr/DIPs*, or their subcellular localization, or in regions/features that are not quantified here.

## Activation of *fru P1* ⋂ *dpr/DIP* neurons results in atypical courtship behaviors

*fru P1* neurons have critical roles in reproductive behaviors. Studies have already determined the function of small subsets of neurons that are responsible for different aspects of behavior (reviewed in *Auer and Benton, 2016*). The *dpr/DIP* tools in hand can further address if additional combinations and/or quantitative differences in the number of *fru P1* neurons are important for behavioral outcomes, given the *fru P1* ⋂ *dpr/DIP* subsets and combinations we examine are distinct from those previously studied. We used the genetic intersectional strategy to selectively activate intersecting neurons by driving expression of TrpA1, a heat-activated cation channel (*Figure 1C*; *von Philipsborn et al., 2011*). This allows for temporal control of neuronal activation by an acute increase of temperature in the courtship chambers (32°C; controls at 20°C). We note that quantitative differences in TrpA1 expression levels may account for some behavioral differences, in addition to the differences in the spatial expression patterns observed (*Figures 2* and *3*).

We find that neuronal activation resulted in decreases in male following and wing extending toward females for several genotypes (*Figure 4* and *Source data 2*). We also observe that neuronal activation of *fru P1* ⋂ *dpr* (13 of 16) and *fru P1* ⋂ *DIP* (2 of 8) genotypes caused atypical courtship behavior toward a female, including double wing extension, and continuous abdominal bending, even if the female had moved away (*Figures 4* and *7*; for abdominal bending phenotype see *Figure 4—video 1*). These atypical behaviors could account for some of the decreases in following and wing extension. For example, if a male is locked into abdominal bending, this would reduce courtship following behavior. Additionally, we found that some males ejaculated on the chamber in five intersecting genotypes: *dpr5* (5 of 15 animals), *dpr9* (3 of 15 animals), *dpr10* (3 of 15 animals), *dpr12* (2 of 15 animals), and *DIP-θ* (4 of 15 animals). Of note, *fru P1* ⋂ *DIP-α* is the only strain that showed a decrease in courtship activities without a concomitant increase in atypical courtship behaviors. This suggests that *fru P1* ⋂ *DIP-α* neurons may normally inhibit courtship behaviors when they are activated.

We next determined if the males require females to reach an arousal threshold needed to perform typical and atypical courtship behaviors, given that several of the courtship behaviors described above occur when the male was not oriented toward the female. To address this question, we examine courtship behaviors in solitary males, using the same temporal activation strategy as above. We find that activation of the *fru P1* ⋂ *dpr/DIP* neurons is sufficient to elicit single wing extension, double wing extension, and abdominal bending in *fru P1* ⋂ *dprs* (11 of 16 genotypes) and *fru P1* ⋂ *DIPs* (3 of 8 genotypes) (*Figures 5* and *7*). Similarly, activating the intersecting *fru P1* neuronal populations of *fru P1* ⋂ *dpr5* (5 of 10 animals), *dpr9* (1 of 10 animals), *dpr10* (1 of 10 animals), *dpr12* (3 of 10 animals), and *DIP-θ* (1of 10 animals) causes males to ejaculate without a female present. Overall, activation of these subsets of *fru P1* neurons is sufficient to direct reproductive behaviors, even if a female is not present, consistent with other neuronal activation experiments (reviewed in *Auer and Benton, 2016*).

## Silencing *fru P1* ⋂ *dpr/DIP* neurons result in courtship changes

Given that activation of *fru P1* ⋂ *dpr/DIP* neuronal subsets resulted in changes in courtship behaviors, we next determine how silencing these neurons impacts male-female courtship, to gain further insight into their roles. To test this, we use the genetic intersectional approach with a *UAS > stop > TNT* transgene (*Figure 1C*; *Stockinger et al., 2005*). The intersecting genotypes express tetanus toxin light chain, which cleaves synaptobrevin, resulting in synaptic inhibition (*Sweeney et al., 1995*). For the control conditions, we also examine courtship behaviors of flies expressing an inactive form of *TNT* (TNTQ), using the genetic intersectional approach.

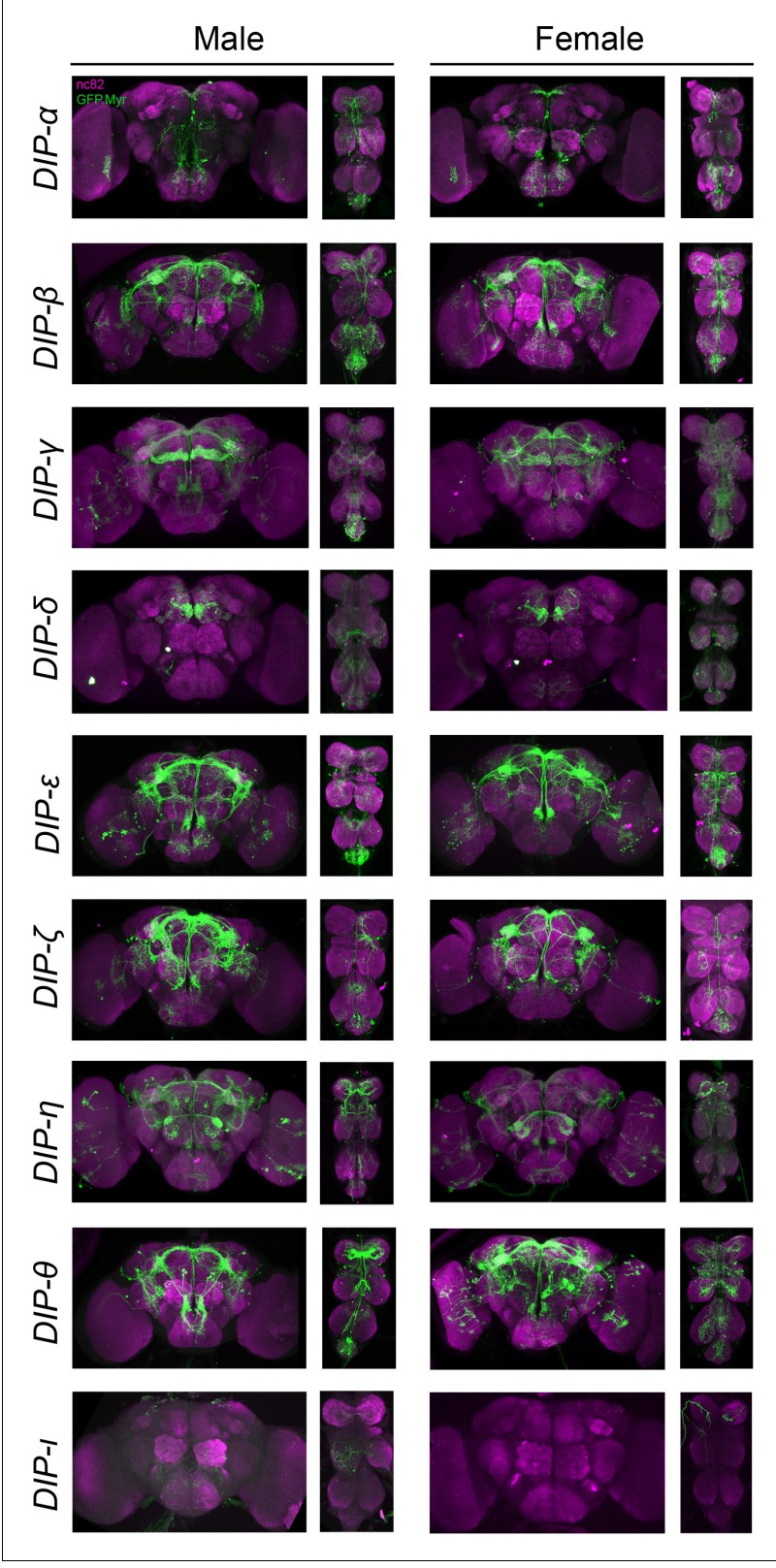

**Figure 3.** Visualization of *fru P1* ∩ *DIP* neurons. Maximum intensity projections of brain and ventral nerve cord tissues from 4- to 7-days old male and female flies. The *fru P1* ∩ *DIP* intersecting neurons are labeled with green (rabbit α-GFP Alexa Flour 488), and neuropil are labeled with magenta (mouse α-nc82, Alexa Flour 633). The genotype is *DIP-Gal4/10xUAS > stop > GFP.Myr; fru P1^{FLP}*, except for *DIP-ι*. This *Gal4* transgenic strains was

*Figure 3 continued on next page*

*Figure 3 continued*

generated using a CRISPR-mediated insertion of the *T2A-Gal4* with the dominant 3xP3-GFP marker. For this strain, *10xUAS > stop > myr::smGdP-cMyc* was used and *fru P1 ∩ DIP* intersecting neurons are labeled with red (rabbit α-Myc, Alexa Flour 568) and then false-colored to green. The neuropil are labeled with magenta (mouse α-nc82, Alexa Flour 633). One Gal4 did not show consistent expression upon intersecting: *DIP-iota*. *DIP-iota* was tested with 10xUAS-RFP, and showed expression outside of *fru P1* neurons.
The online version of this article includes the following figure supplement(s) for figure 3:

**Figure supplement 1.** Visualization of *fru P1 ∩ DIP* neurons from 0- to 24-hr-adults.

In addition to scoring courtship behaviors, overall motor defects were also scored (*Figure 6*). Given that neuronal silencing in several genotypes resulted in motor defects, in which the male fell and was unable to quickly right himself ($p<0.005$ for strong motor defects; $0.05>p>0.005$ for minor motor defects in *Figure 6*), we quantify the time when the fly could not right himself as 'motor defect' and subtracted this from the overall courtship time for behavioral indices (*Figure 6*). The intersecting genotypes that consistently demonstrate motor defects additionally show decreases in following and wing extension upon silencing, likely due to some motor defect (*fru P1 ∩ dpr1, dpr3, dpr4, dpr5, dpr9, dpr10, dpr11, dpr12, dpr15*, and *DIP-η*). Additional courtship behavioral indices and latencies were quantified and those with motor defects show additional courtship phenotypes (*Source data 2*). However, seven intersecting genotypes had a decrease in following/wing extension indices and only minor or no motor defect (*fru P1 ∩ dpr2, dpr6, dpr17, dpr18, DIP-ε, DIP-θ,* and *DIP-γ*). One genotype, *fru P1 ∩ dpr7*, has an increase in following/wing extension with neuronal silencing. In the case of *fru P1 ∩ dpr7*, we do not detect GFP expression in the central or peripheral nervous system in adults, so the neurons underlying this phenotype remain to be determined.

The seven intersecting genotypes with no or minor motor defects, in the courtship assay, were further analyzed for locomotor defects using the *Drosophila* Activity Monitor (DAM) Assay (Trikinetics; *Source data 2*), along with genotypes *fru P1 ∩ dpr7*, and *fru P1 ∩ dpr10*, which has a strong motor defect in the courtship assay. If there is a significant difference, the intersecting genotype with neuronal silencing has increased locomotor activity in the DAM assay, suggesting that the courtship phenotypes are not due to overall loss in locomotor activity.

As above in the neuronal activating experiments, silencing *fru P1 ∩ dprs* (13 of 17 genotypes) is more likely to cause a courtship defect than silencing *fru P1 ∩ DIPs* (4 of 9 genotypes). Given the large effect size of the courtship defects compared to the smaller effect size of the motor defect, it is clear that silencing *fru P1 ∩ dpr/DIP* neurons in the central nervous system, for most genotypes, suppresses courtship (*Figure 6*). This is consistent with previous studies that have found that silencing *fru P1* neurons in males leads to decreased courtship toward a female (*Manoli et al., 2005*; *Stockinger et al., 2005*). Interestingly, *fru P1 ∩ DIP-α* is the only strain to demonstrate motor defects, but no change in courtship behaviors upon silencing, underscoring the previous hypothesis that these neurons may normally be inhibitory for courtship.

## Meta-analysis of male *fru P1 ∩ dpr/DIP* expression patterns and behavioral data

Next, we determine if intersecting genotypes with similar expression patterns also have similar behavioral outcomes in the neuronal activating and silencing experiments described above. This allows us to determine if behavioral phenotypes are most often seen when there are large numbers of neurons activated, or when certain combinations of neurons are activated. We used a heuristic approach and generated a heatmap that groups *dprs* and *DIPs* based on similarity of the *fru P1 ∩ dpr/DIP* membrane-bound-GFP expression data (*Figure 7A*). At the top of the heatmap is a dendrogram showing the relationships in expression data, grouping those that are most similar together (from data in *Source data 1*). The bottom has colored dots that indicate the behavioral changes observed in the three different behavioral perturbation data sets. The scoring key for the GFP expression phenotypes is shown (*Figure 7B* and *Source data 1*). Only the 24 intersecting genotypes with quantified GFP expression data are included in the heatmap.

There is a set of eight intersecting genotypes grouped together on the right of the dendrogram that all have expression in the mushroom body and several regions within the lateral protocerebral

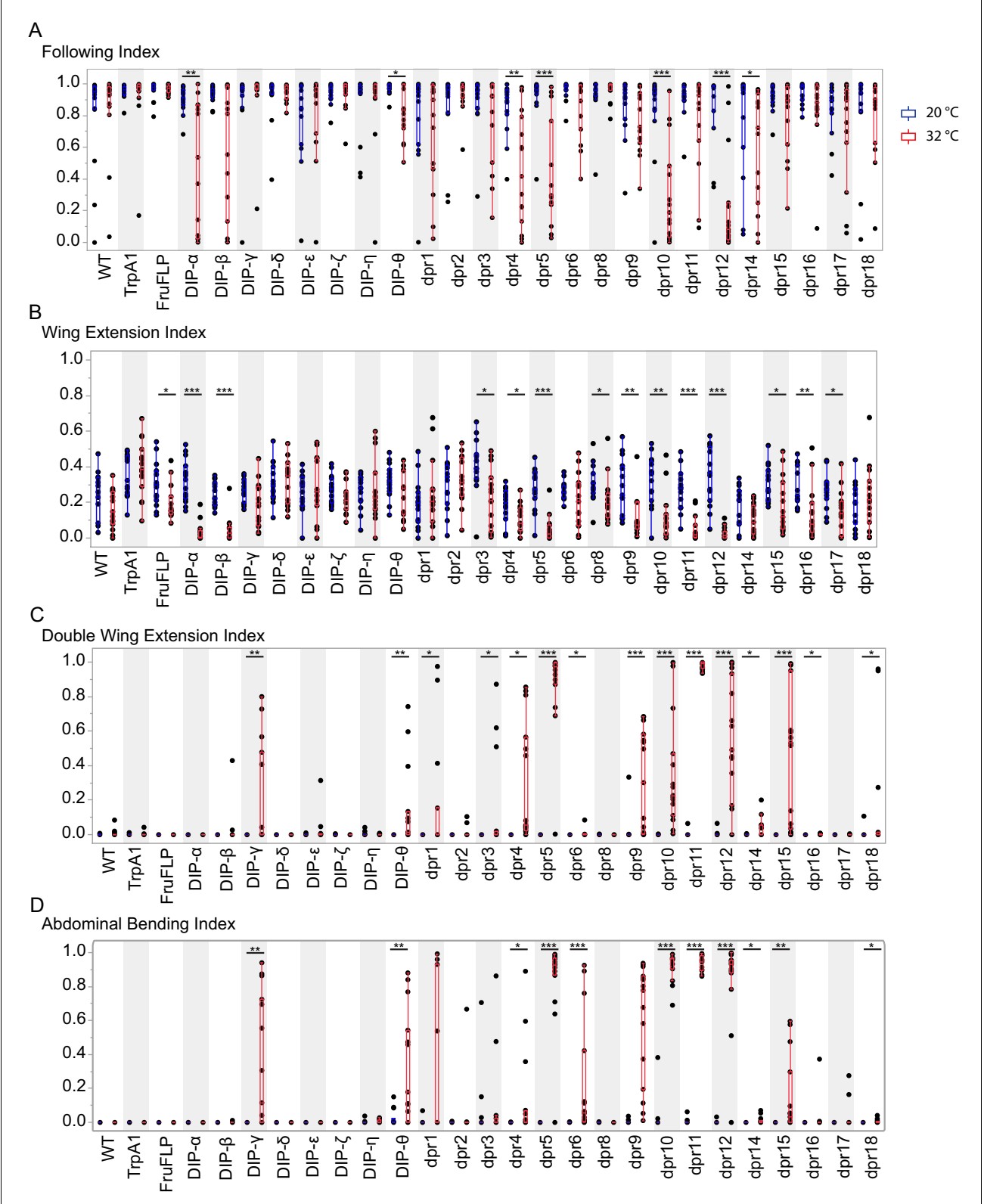

**Figure 4.** Activation of *fru P1* ⋂ *dpr/DIP* intersecting neurons results in atypical courtship behaviors. Courtship behaviors of *dpr/DIP-Gal4/ UAS > stop > TrpA1; fru P1ᶠᴸᴾ* males were recorded at the control temperature (20°C, blue box plots) and the activating temperature for TrpA1 (32°C, red box plots). The control genotypes are the wild-type strain Canton S, and the *UAS > stop > TrpA1* and *fru P1ᶠᴸᴾ* single transgenes, which were crossed to Canton S. Virgin Canton S (*white*) females were used as targets. (**A**) Following index is the fraction of time the male spent oriented toward or chasing

*Figure 4 continued on next page*

*Figure 4 continued*

the female around the chamber. (B) Wing extension index is the fraction of time the male spent unilaterally extending and vibrating his wing. (C) Double wing extension index is the fraction of time the male spent extending and vibrating both wings simultaneously. (D) Abdominal bending index is the fraction of time the male spent curling his abdomen under. A movie is provided to show a male with atypical abdominal bending (*fru P1 ∩ dpr1* genotype; *Figure 4—video 1*). The lines on the quantile box plot correspond to the quantiles in the distribution output, with the center line as the median. The whiskers extend from the 1st and 3rd quartiles to the edges, which correspond to the minimum and maximum values, excluding outliers. The nonparametric Wilcoxon rank sum test was used to test for significant difference between control and activating temperature within each genotype. n = 15. *p<0.05, **p<0.005, ***p<0.0005. All lines were examined for expression of TrpA1 to confirm the system was working effectively (data not shown).

The online version of this article includes the following video for figure 4:

**Figure 4—video 1.** A movie showing a male that has atypical abdominal bending when *fru P1 ∩ dpr1* neurons are activated by TrpA1.
https://elifesciences.org/articles/63101#fig4video1

complex, but varied expression across the other morphological features (*Figure 7A*; *fru P1 ∩ dpr4, dpr5, dpr8, dpr9, dpr10, dpr12, dpr14,* and *DIP-γ*). Seven have similar types of atypical courtship behaviors in the activating experiments (excluding *fru P1 ∩ dpr8*), in the male-female courtship assays. These seven also have similar behavioral phenotypes in the male-alone condition, indicating that the activation threshold in these lines can be achieved without a female present (*Figure 5*).

Furthermore, among the eight genotypes, there are four intersecting genotypes that have male ejaculates in the chamber, in both the male-female and male-alone neuronal activation assays. Among the eight intersecting genotypes, the four with the male ejaculates have the highest cell body counts in the abdominal ganglion, a region in the ventral nerve cord that has previously been shown to drive ejaculation (*Source data 1*; *Tayler et al., 2012*). However, not all intersecting genotypes with expression in the abdominal ganglion show the ejaculation phenotype, even those with a relatively high number of neuronal cell body counts in the abdominal ganglion (see *fru P1 ∩ dpr1 and dpr2*), suggesting that it is the combination of neurons that are activated that is critical for this phenotype. Furthermore, there is an intersecting genotype that does not have mushroom body expression, but also has the ejaculation phenotype (*fru P1 ∩ DIP-θ*). These results reveal how different combinations and numbers of neurons can direct a similar behavioral outcome. Overall, the results point to a critical role for interactions between the mushroom body and protocerebral complex in directing courtship behaviors, which are modified by being activated in combination with other neuronal populations. This is consistent with an idea put forth previously that posited connections between these two brain regions may integrate diverse external stimuli with internal physiological state and previous behavioral experience (*Yu et al., 2010*).

Twenty-two intersecting genotypes have expression in different regions of lateral protocerebral complex, but no consistent expression in the mushroom body. An examination of the behavioral data reveals no phenotypes correlated with the lateral protocerebral complex expression data. While the lateral protocerebral complex is critical for higher order processing, the data further supports the idea that interactions across different combinations of activated neurons, in each intersecting genotype, are critical for the behavioral outcomes and underscores how different patterns of neuronal activity can direct similar behavioral outcomes.

## Correlation of *fru P1 ∩ dpr/DIP* expression patterns

As an additional heuristic tool, we plot the correlation of the GFP expression patterns for the male and female data (*Figure 7C*). One goal is to gain insight into whether Dprs with the same DIP interacting partners are co-expressed in the same regions of the central nervous system. This allows us to gain understanding into the mechanisms used by these IgSF molecules to direct cell adhesion. Another goal is to determine if protein-protein interactions may occur through *cis* (within the same neuron) vs *trans* (across neurons) interactions. For example, if protein-protein interactions are in *trans*, then the Dpr/DIP interacting partners will be expressed in different neurons and may not have correlated expression patterns. Experimental evidence suggests *trans* interaction are functionally important, but there is currently no evidence that interactions in *cis* are important (*Xu et al., 2018*; *Ashley et al., 2019*; *Courgeon and Desplan, 2019*; *Menon et al., 2019*; *Venkatasubramanian et al., 2019*; *Xu et al., 2019*). To address these questions, the plots are

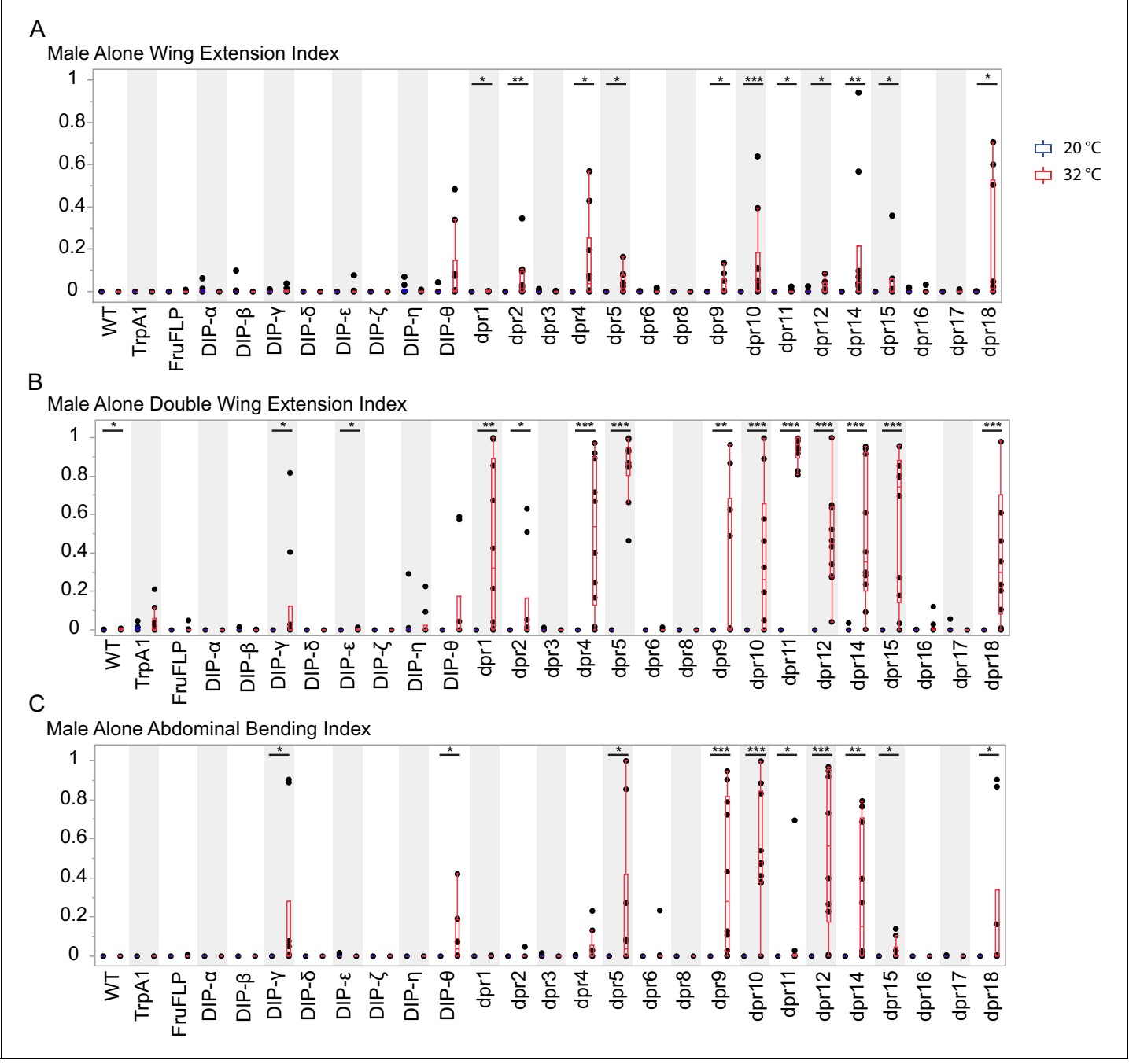

**Figure 5.** Activation of *fru P1 ∩ dpr/DIP* intersecting neurons is sufficient to induce courtship behaviors in solitary males. Courtship behaviors of *dpr/DIP-Gal4/ UAS > stop > TrpA1; fru P1^FLP* solitary males were recorded at the control temperature (20°C, blue box plots) and the activating temperature (32°C, red box plots). The control genotypes are the wild type strain Canton S, and the *UAS > stop > TrpA1* and *fru P1^FLP* single transgenes, which were crossed to Canton S. (A) Wing extension index, (B) Double wing extension index (C) Abdominal bending index, and quantile box plots are as described in *Figure 3*. The nonparametric Wilcoxon rank sum test was used to test for significant difference between control and activating temperature within each genotype. n = 10. *p<0.05, **p<0.005, ***p<0.0005.

annotated with DIPs (colored dots) that each Dpr interacts with on the right (based on interactome from *Cosmanescu et al., 2018*).

It appears that some Dprs/DIPs that bind the same partner have the most similar expression patterns. For example, in males *fru P1 ∩ dpr1* and *dpr2* have highly correlated expression and both Dpr1 and Dpr2 interact with DIP-η, DIP- ι, and DIP- κ. In addition, the male *fru P1 ∩ DIP- η*

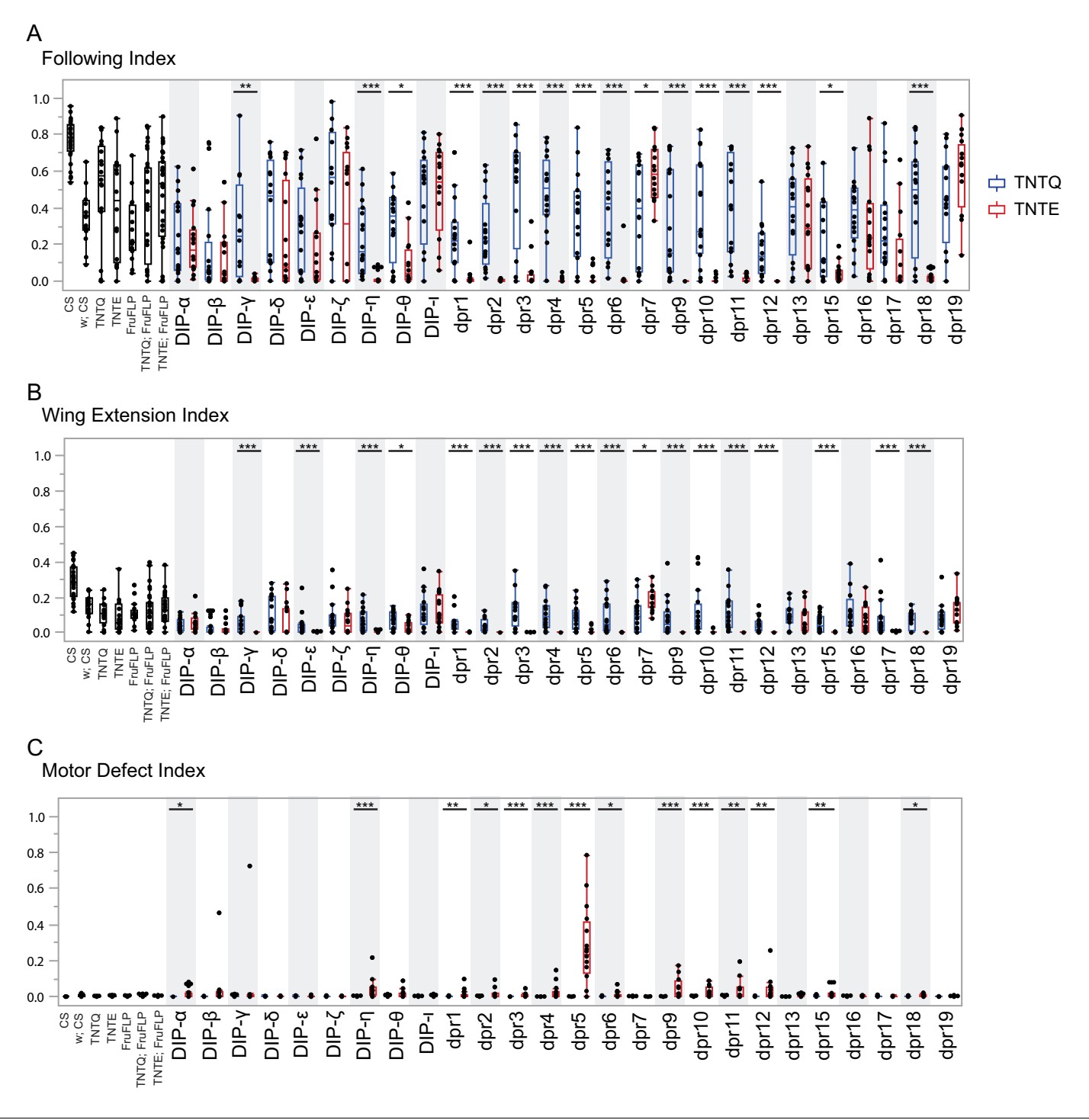

**Figure 6.** Silencing *fru P1* ∩ *dpr/DIP* intersecting neurons results in atypical courtship and severe motor defects. Courtship behaviors of *dpr/DIP-Gal4/ UAS > stop > TNTQ; fru P1FLP* (control condition, blue boxplots) and of *dpr/DIP-Gal4/ UAS > stop > TNTE; fru P1FLP* (experimental condition, red boxplots) males were quantified. Control genotypes (black boxplots) are the wild-type strain Canton S and Canton S (*white*), *fru P1FLP*, *UAS > stop > TNTQ*, and *UAS > stop > TNTE* single transgenes, as well as *UAS > stop > TNTQ; fru P1FLP* and *UAS > stop > TNTE; fru P1FLP* double transgenes. The single and double transgene controls were crossed to Canton S (*white*). The *dpr-* or *DIP- Gal4* is listed on the x-axis and the fraction of time spent performing the behavior is on the y-axis. (**A**) Following index, (**B**) wing extension index, and the quantile box plots are as described in *Figure 4*. (**C**) Motor defect index is the fraction of time the fly spent on his back after falling. The nonparametric Wilcoxon rank sum test was performed to determine significant differences between experimental and control conditions with the same *dpr/DIP-Gal4*. n = 16 for all genotypes except for Canton S, and the

*Figure 6 continued on next page*

*Figure 6 continued*

double transgene controls, which have n = 32. Those three genotypes were assayed twice, n = 16 each time, to ensure consistency throughout the duration of the experiment and pooled for this analysis. *fru P1* ∩ *DIP- ι* and *dpr7* were not included in the activating experiments, but are here, as those lines were not available until after completion of the activation assays. The *dpr19-Gal4* did not produce an expression pattern in the nervous system, using both a *10XUAS-RFP* reporter and the intersectional approach, at the time points examined. n = 16. *p<0.05, **p<0.005, ***p<0.0005.

expression pattern is highly correlated with *fru P1* ∩ *dpr1*, and *dpr2.* Similarly, in females, *fru P1* ∩ *dpr1*, *dpr2*, and *dpr3* have highly correlated expression, with Dpr1, Dpr2, and Dpr3 interacting with DIP-η and DIP- ι, and Dpr1 and Dpr2 interacting with DIP- κ. Therefore, in both males and females, *fru P1* neurons may express a reporter for both a *DIP* and a *dpr* that can bind to it. In both males and females *fru P1* ∩ *dpr11* has correlated expression with *fru P1* ∩ *dpr1* and *dpr2,* but Dpr11 interacts with DIPs that Dpr1, and Dpr2 do not interact with. This suggests that expression of different combinations of Dprs that interact with different DIPs is a mechanism used to direct the specificity and strength of interactions. *fru P1* ∩ *DIP-α* and *DIP-δ* have the most restricted expression patterns and they are not highly correlated with the expression patterns of their interacting Dpr partners, in either males or females. Consistent with the single-cell RNA-seq analysis (below), these data show that reporters for Dprs that bind the same DIP can be expressed in the same neurons, as well as in different neurons. Additionally, some Dprs with different binding partners have correlated expression, which could be a mechanism to mediate the specificity and strength of neuronal adhesion.

## Single-cell mRNA sequencing analysis of male *fru P1* neurons

To determine the repertoires of *dprs/DIPs* expressed in individual *fru P1* neurons, we perform single-cell RNA-sequencing (10X Genomics platform). The analysis was performed on male central nervous system tissues (48 hr pupal stage), from flies expressing membrane-bound GFP in *fru P1* neurons. We chose this stage to gain further insight into how the *dprs/DIPs* direct development of *fru P1* neurons, as this is the stage where Fru$^M$ has peak expression (~1,700 Fru$^M$ neruons, *Lee et al., 2000*). The data were filtered to include only the *fru P1* neurons, based on detection of the membrane-bound GFP mRNA, resulting in 5621 neurons. We find that all *fru P1* neurons express at least one *dpr/DIP*. A principal component analysis (PCA) was performed using only *dpr/DIP* gene expression values, and the dimensionality was reduced with the UMAP algorithm (*McInnes et al., 2018* and *Stuart et al., 2019*). Cells with similar *dpr/DIP* expression will cluster closely with one another in the UMAP plot (*Figure 8A*). A visual inspection of the UMAP plot reveals that the patterns of *dpr/DIP* expression are not distinct enough to generate clusters that have large separation in the UMAP plot. We note that previous single-cell RNA-seq studies from adult central nervous system tissues found that *dprs/DIPs* are cell identity marker genes (*Croset et al., 2018*; *Davie et al., 2018*; *Allen et al., 2020*), with higher, enriched expression in subsets of neurons, rather than pan-neuronal expression, consistent with our results.

For each cluster, we next determine if a combination of *dprs/DIPs* are responsible for each cluster identity. We examine the average expression and the percent of cells with expression of each *dpr/DIP* in each cluster (*Figure 8B*). The UMAP clustering patterning is not due to co-expression of Dpr/DIP interacting partners (*Figure 8C*), based on a visual inspection, which might be expected if Dprs/DIPs predominantly interact in *cis*. We find that the majority of the *DIPs* have high average expression in one cluster, with a large percent of the cells in the cluster having expression. This is distinct from the majority of the *dprs*, where the average expression and the percent of cells that express the *dpr* is moderate and similar across many clusters. Furthermore, the distribution of the expression patterns overlaid on the UMAP plot for each *dpr/DIP* shows that the *DIP*s have more restricted expression. For example, *dpr21* is broadly detected across the UMAP plot, whereas *DIP-α* has restricted expression in cluster 10, at the upper left-hand side of the UMAP plot (subset of expression patterns in *Figure 8D*; for all *dprs/DIPs* see *Figure 8—figure supplement 1*). This observation is consistent with cell-type-specific RNA-seq data from the *Drosophila* visual system (*Tan et al., 2015*; *Cosmanescu et al., 2018*; *Davis et al., 2020*), where the *DIPs* were among the most specifically expressed genes and *dprs* were more broadly expressed. This suggests that DIPs may have different functional roles, compared to Dprs, in terms of directing synaptic specificity or cell adhesion properties of the neuron.

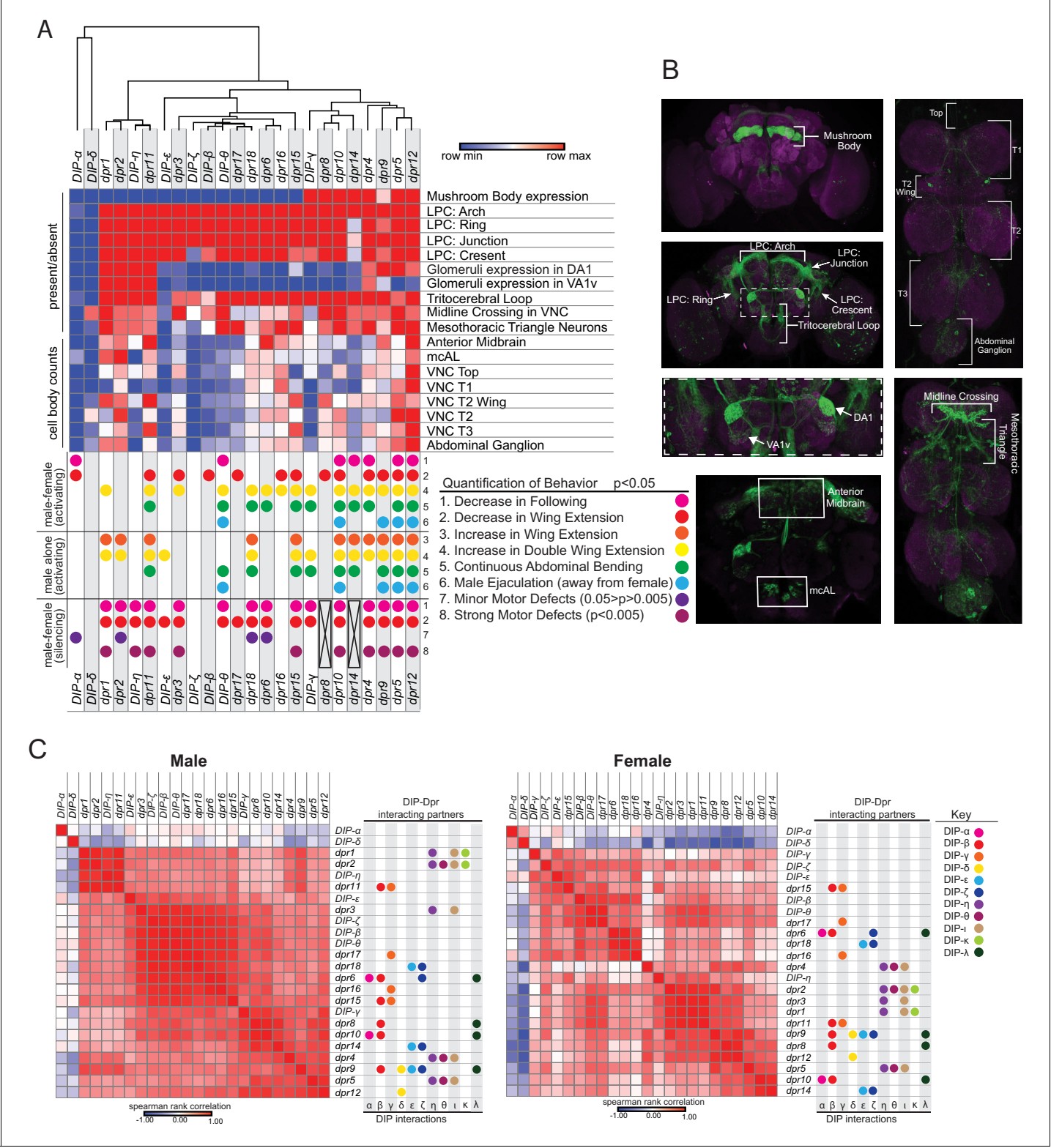

**Figure 7.** Meta-analysis of expression patterns of *fru P1* ∩ *dpr/DIP* intersecting neurons and behavior data. Meta-analysis using behavior data and image analysis data of 4- to 7-day-old flies. (**A**) Heatmap of *fru P1* ∩ *dpr/DIP* intersecting neurons expression patterns in the male adult CNS. For each row, the minimum (blue), middle (white), and maximum (red) values are indicated. The top of the heatmap shows the relationship across the expression patterns of the *dprs* and *DIPs*, with a dendrogram. The summary of phenotypic analyses of male sexual behaviors, using either activating or silencing effector genes (see *Figures 4–6*), is shown below the heatmap. The dot indicates a significant change in behavior (p<0.05, unless indicated). The black

*Figure 7 continued on next page*

Figure 7 continued

X indicates that there was no experimental progeny from the cross, due to lethality, and therefore were not tested behaviorally. (B) Labeled confocal images showing the morphological featured scored. (C) Correlation analysis of GFP expression results (male on left and female on right). The scale for the spearman correlation is −1 (blue) to 1 (red). The dots to the right indicate the DIP interacting partners for each Dpr (left-hand side of each graph) (Dpr-DIP interactome based on *Cosmanescu et al., 2018*). The full data set is provided (*Source data 1*).

We also generated a dendrogram by hierarchical clustering to visualize which *dprs* and *DIPs* have the most similar expression patterns across the *fru P1* neurons (*Figure 8—figure supplement 2*). We find that some *dprs* and *DIPs* that have shared interacting partners have the most similar expression to each other. This includes the following pairs: *dpr2* and *dpr3*; *dpr6* and *dpr10*; *dpr16* and *dpr17*; and *DIP-ζ* and *DIP-ε*. We find that *DIP-α*, and *DIP-ι* have the most distinct expression patterns from the rest of the interactome, which may be due to the low number of cells in which they are detected (see Upset plot described below; *Source data 3*). For some neurons, co-expression of *dprs* and *DIPs* with the same interacting partners may be a mechanism to generate different adhesion properties, as noted above.

Next, we examine the number of different combinations of *dpr* and *DIP* expression repertoires. A *dpr/DIP* gene is considered expressed if the normalized and scaled expression data value is >1, excluding those that may have stochastic expression detection due to low expression levels (5218 neurons remain). There are 458 neurons that express only one *dpr* or *DIP*. The range of neurons that express 2–8 *dprs* and *DIPs* is between 451–653 neurons (4024 neurons total); that express 9–11 *dprs* and *DIPs* is between 105–332 neurons (657 neurons total); and that express 12–15 *dprs* and *DIPs* is

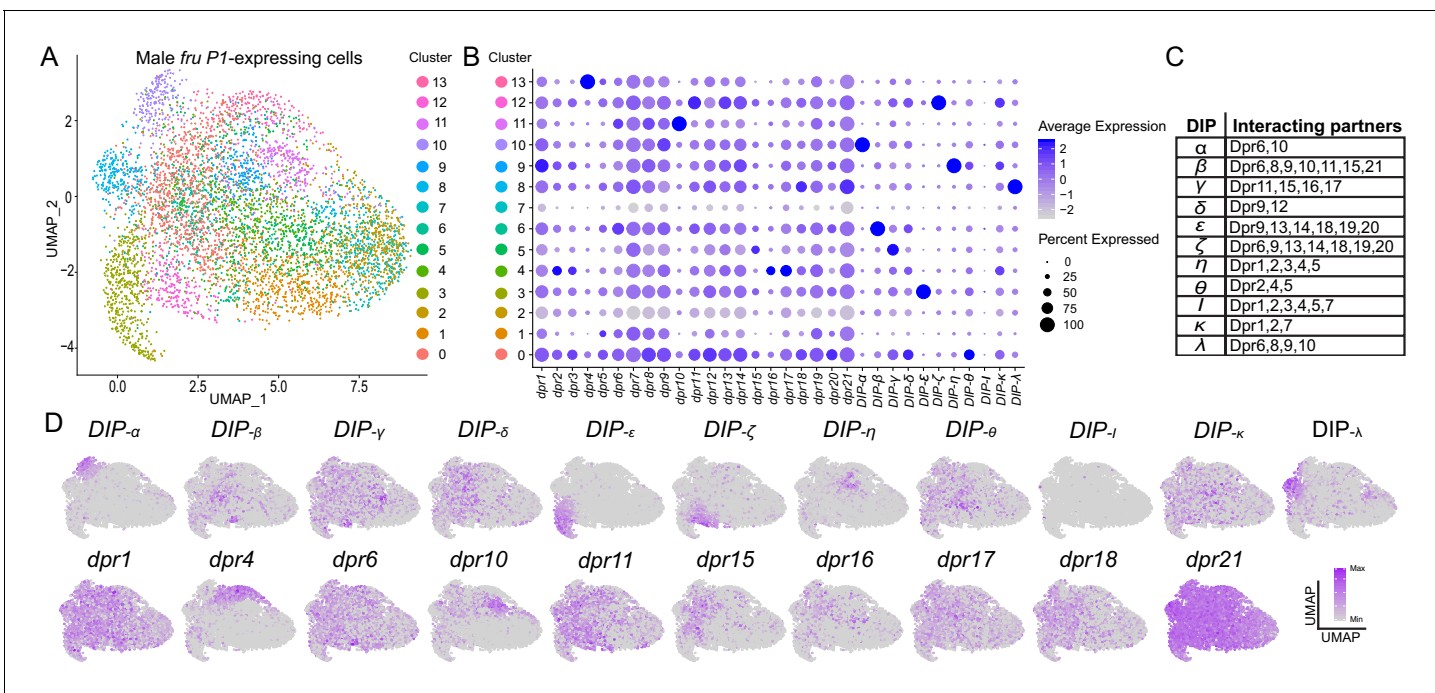

**Figure 8.** Single-cell RNA-sequencing analysis of *dpr/DIP* expression analysis in male *fru P1* neurons. (A) UMAP plot of 5621 *fru P1* neurons, isolated from male tissue 48-hr after puparium formation. The data are clustered on *dpr/DIP* gene expression. (B) Dot plot showing the expression of *dpr/DIP* genes across all clusters identified in UMAP. Dot diameter indicates the fraction of cells expressing each gene in each cluster, as shown in legend. Color intensity indicates the average normalized expression levels. (C) Heterophilic interactions that have been demonstrated between DIPs and Dprs (*Cosmanescu et al., 2018*). (D) A subset of expression visualization of *DIP*s (top row) and subset of *dpr*s (bottom row) in the UMAP-clustered cells. *dpr* or *DIP*-positive cells are labeled purple and color intensity is proportional to log normalized expression level shown in legend. The UMAP visualization for all *dprs/DIPs* is provided (*Figure 8—figure supplement 1*). The numerical expression values are in *Source data 3*.

The online version of this article includes the following figure supplement(s) for figure 8:

**Figure supplement 1.** *dpr/DIP* expression visualization in the UMAP cluster.
**Figure supplement 2.** Hierarchical cluster dendrogram of *dpr/DIP* expression.

between 5–45 neurons (79 neurons total; *Source data 3*). Next, we look at the number of neurons with the same expression repertoire. This can be ascertained using an 'Upset' plot, which is conceptually similar to a Venn Diagram, but accommodates a large number of conditions. Here, there are the 5218 single neuron expression conditions. The majority of expression repertoires that are detected in more than one neuron are those for which the neuron only expresses one *dpr* or *DIP* (single dots on bottom of Upset Plot, 457 neurons; *Source data 3*). There were also 466 neurons that had shared co-expression combinations due to expression of 2–5 *dprs* and *DIPs*. The majority of *fru P1* neurons had a unique repertoire of *dpr/DIP* expression (4295 neurons), due to expression of different combinations of 2–15 *dprs* and *DIPs* (*Source data 3*). In developing *fru P1* neurons, this range of expression patterns of the *dprs* and *DIPs* may provide a mechanism to generate different connectivity properties for each neuron.

## A higher resolution analysis of *fru P1* ∩ *DIP-α* reveals additional sexually dimorphic expression patterns

After examining all the *fru P1* ∩ *dpr/DIP* patterns, and the single-cell RNA-seq data, it became apparent that *fru P1* ∩ *DIP-α* neurons are sexually dimorphic, and that this is one of the genotypes with the fewest cells among the genotypes that were scored (*Source data 1*). Additionally, the *fru P1* ∩ *DIP-α* neurons have arborization patterns that facilitate analysis of sex-differences in fine-scale processes that would be obscured in intersecting genotypes with broad expression. There are sex-differences in the superior medial protocerebrum region (SMP; *Figure 9A and B*, subpanels I), where females have a longer (dotted-line) and broader projection (arrowhead), as compared to males. Moreover, in the medial part of the midbrain, an 'M' shaped peak forms ('M'-like) in males that is not typically observed in females (curved dotted-line, *Figure 9A and B*, subpanels II and III). Additionally, in the ventral lateral protocerebrum region (VLP) there are neuronal cell bodies (arrowhead, *Figure 9A and B*, subpanels II and III), and projections in a 'square' shaped pattern that are more frequently observed in females (closed dotted-line, *Figure 9A and B*, subpanels II and III). There is also a greater frequency of neuronal cell bodies present in the subesophageal ganglion (SEG) in females, as compared to males (arrowhead, *Figure 9A and B*, subpanels IV). In the abdominal ganglia (AbG) of the ventral nerve cord there is a higher density of projections in males (*Figure 9A and B*, subpanels V). In contrast, females have a distinct 'forceps' shaped pattern in the AbG region (arrowhead, *Figure 9A and B*, subpanels V). Taken together, it appears that the sex differences are due to differences in the number of neurons and also in the morphology of projections and arborizations (*Figure 9*).

## Changing the sex of *DIP-α* neurons alters the *fru P1* ∩ *DIP-α* co-expressing patterns

We next investigated whether perturbations of the sex hierarchy genes impact fine-scale sex differences in *fru P1* ∩ *DIP-α* neurons (*Figure 9* and *Source data 4*). In this screen, *DIP-α-Gal4* drives broad expression of each transgene (see *Source data 4* for *DIP-α-Gal4* expression), and the *fru P1* ∩ *DIP-α* patterns are visualized. First, we examine the phenotypes when we overexpressed the female-isoform of the sex hierarchy gene *tra* (*tra^F^*). This is expected to feminize the neurons by switching to female-specific splicing of *fru P1* (*Figure 1*). In males, the projections in the SMP became more female-like (*Figure 9C–D*, subpanels I). In the medial part of midbrain, the horizontal projections in half of the male samples were either more female-like or not detected (*Figure 9C–D*, subpanels II). Similarly, among half of the male samples, the neuronal patterns within the AbG are either more female-like or absent (*Figure 9C–D*, subpanels V). We observed unexpected phenotypes in females upon overexpressing Tra^F^, which suggests that quantitative differences in Tra^F^ have biological outcomes, as we previously suggested (*Arbeitman et al., 2016*). For instance, a lateral ascending neuronal projection was observed more frequently in the VLP region (*Figure 9C–D*, subpanels III dotted line). However, the neuronal cell bodies in the VLP, the adjacent 'square' shaped projection patterns (closed dotted-line, *Figure 9C–D*, subpanels III) and the medial horizontal projection (*Figure 9C–D*, subpanels III) were less frequently observed, as compared to control females.

We also examined phenotypes after Fru^M^ over-expression, by driving broad expression in *DIP-α* cells and visualizing the *fru P1* ∩ *DIP-α* neurons. We tested three isoforms of Fru^M^ (*UAS-Fru^MA^*, *UAS-Fru^MB^*, and *UAS-Fru^MC^*) and found that they could effectively produce Fru^M^ in the expected *DIP-α*

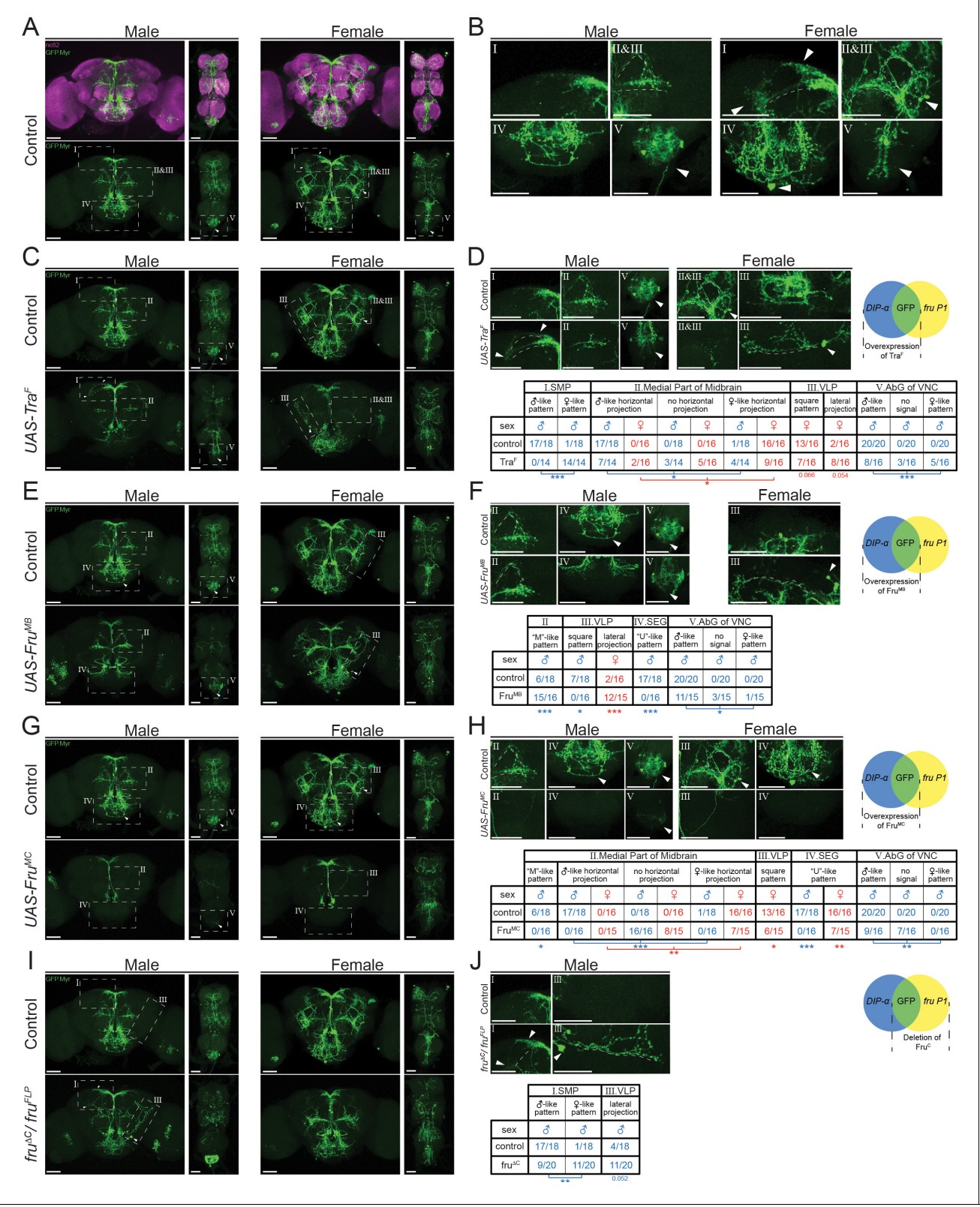

**Figure 9.** Higher resolution analyses of *fru P1* ∩ *DIP-α* neurons with sex hierarchy perturbations. Confocal maximum intensity projections of brains and ventral nerve cords from 4- to 7-day-old adult flies. *fru P1* ∩ *DIP-α* neurons are in green (rabbit α-GFP Alexa Flour 488). Staining with the α-nc82 neuropil maker shows brain morphology in magenta (mouse α-nc82, goat α-mouse Alexa Flour 633). Image data were captured with ×20 objective, with scale bars showing 50 μM (A-J). Higher magnification images were generated using the Zeiss Zen software package (B, D, F, H, and J). Roman

*Figure 9 continued on next page*

**Figure 9 continued**

numerals are consistent across the panels in the same row. Venn diagrams show where membrane-bound GFP and sex hierarchy transgenes are expressed. (A) *fru P1* ∩ *DIP-α* expression patterns in males and females. (B) Computationally magnified images, with sexually dimorphic regions indicated. Subpanels show: [I] superior medial protocerebrum (SMP) region of the brain; [II and III] medial part of midbrain region, where there are horizontal projections, and the 'M'-like pattern (more frequent in males). The square pattern (more frequent in females) is in the ventral lateral protocerebrum (VLP) region of the brain. The medial horizontal projection is in a more exterior section of the confocal stack then the other features [II and III]; [IV] Subesophageal ganglion region of the brain (SEG). The U-like pattern and a set of cell bodies more frequently found in females are shown; [V] The abdominal ganglion of the ventral nerve cord (AbG). (C-J) Examination of morphology of *fru P1* ∩ *DIP-α* neurons when sex hierarchy transgenes are expressed in *DIP-α* neurons. The quantification and statistics are provided in a table within the subpanel on the right of each row. This figure only shows regions that had significant changes due to sex hierarchy perturbation (full dataset provided; *Source data 4*). (C-D) Tra$^F$ overexpression in *DIP-α* neurons. [III] a lateral projection in females that is not shown in wild-type data in panel B. (E-F) Fru$^{MB}$ overexpression in *DIP-α* neurons. (G-H) Fru$^{MC}$ overexpression in *DIP-α* neurons, (I-J) Fru$^C$ isoform deletion. Fru$^{MC}$ is absent or highly reduced in *fru P1* neurons in this genotype, as transheterozygous for *fru$^{FLP}$/ fru$^{ΔC}$*. Statistical significance of the differences in morphological features, between same sex control and genotypes with sex hierarchy transgene expression are indicated. Comparisons were done using the Fisher's exact test (*p<0.05, **p<0.005, ***p<0.0005). The morphological features with significant differences are indicated by lines below the table (male in blue and female in red). n ≥ 15 for each category. The genotypes of the samples shown are: *DIP-α$^{Gal4}$; UAS>stop>GFP.Myr/+; fru$^{FLP}$/+* (A-B), *DIP-α$^{Gal4}$; UAS>stop>GFP.Myr/ UAS-Tra$^F$; fru$^{FLP}$/+* (C-D), *DIP-α$^{Gal4}$; UAS>stop>GFP.Myr/ UAS-Fru$^{MB}$; fru$^{FLP}$/+* (E-F), *DIP-α$^{Gal4}$; UAS>stop>GFP.Myr/ UAS-Fru$^{MC}$; fru$^{FLP}$/+* (G-H), *DIP-α$^{Gal4}$; UAS>stop>GFP.Myr/+; fru$^{FLP}$/ fru$^{ΔC}$* (I-J). Brain region nomenclature are consistent with previous reports (*Insect Brain Name Working Group et al., 2014*).

The online version of this article includes the following figure supplement(s) for figure 9:

**Figure supplement 1.** Sex hierarchy perturbation in only *fru P1* ∩ *DIP-α* neurons.

pattern (*Source data 4*). Overexpression of Fru$^{MB}$ and Fru$^{MC}$ had large phenotypic impacts, whereas Fru$^{MA}$ did not, consistent with previous functional studies of the Fru$^M$ isoforms (*Nojima et al., 2014*; *von Philipsborn et al., 2014*). Overexpression of Fru$^{MB}$ resulted in a higher frequency of the 'M' shaped projection pattern in males (curved dotted-line, 'M'-like, *Figure 9E–F*, subpanels II), while the 'U' shaped SEG projection was not observed as frequently ('U'-like, *Figure 9E–F*, subpanels IV). The density of the neuronal projections in the AbG was also reduced. In females, the lateral ascending neuronal projection in the VLP region is observed more frequently (*Figure 9E–F,III*). The overexpression of Fru$^{MC}$ led to substantial reduction of *fru P1* ∩ *DIP-α* intersecting neurons in both males and females (*Figure 9G–H*, subpanels III), which could be due to a loss of neurons and/or their projections. The phenotype could also be due to reduced *DIP-α-Gal4* expression, given overexpression of Fru$^M$ was previously shown to reduce expression of some IgSFs (*Dalton et al., 2013*).

A loss of the Fru$^{MC}$ isoform, only in *fru P1* neurons, had less strong phenotypic consequences (*fru$^{FLP}$/ fru$^{ΔC}$*; *Figure 9I–J*). In males, the SMP projections appeared more female-like and there was an increase in neurons with a lateral projection, due to loss of the Fru$^{MC}$ isoform. Therefore, overexpressing Fru$^{MC}$ isoform in the broad *DIP-α-Gal4* pattern impacts *fru P1* ∩ *DIP-α* neurons more substantially than loss of Fru$^{MC}$ isoform in only *fru P1* neurons. This suggests that the wildtype Fru$^{MC}$ spatial expression pattern is critical for function. It is also possible that overexpression of Fru$^{MC}$ interferes with other Fru isoform functions, since all Fru isoforms have a BTB domain thought to mediate dimerization. Furthermore, overexpression could titrate out co-transcriptional regulators leading to larger impacts. If we limit the overexpression of Tra$^F$ and Fru$^M$ to only *fru P1* ∩ *DIP-α* neurons using an additional transgene (*tub>GAL80>*), we also see phenotypes that are less severe than observed when overexpression is in all *DIP-α* neurons (see *Figure 9—figure supplement 1*). Overall, quantitative and spatial changes in the expression of sex hierarchy genes alters the sexually dimorphic *fru P1* ∩ *DIP-α* patterns. This demonstrates that sex differences in morphology are downstream of sex hierarchy regulation, through both cell autonomous and non-autonomous mechanisms.

## Knockdown of *DIP-ε* in *fru P1* ∩ *DIP-α* co-expressing neurons alters the expression patterns

To determine the functional roles of *dprs/DIPs* in *fru P1* neurons, we conducted an RNAi and overexpressor screen. We use the *DIP-α* and *DIP-δ* drivers, given that they have the most restricted intersecting expression patterns, with the fewest neuronal cell bodies among the genotypes scored (*Source data 1*), which facilitates visually identifying altered patterns in *fru P1* ∩ *DIP* neurons, as discussed above. Here, the *DIP-Gal4* drives expression of an RNAi or over-expressor transgene of other *dprs/DIPs*. It should be noted that while these *fru P1* ∩ *DIP* intersecting patterns are highly

restricted, the *DIP-Gal4* patterns that drive the perturbation are broader. Out of the 36 genotypes screened, only one perturbation robustly alters the *fru P1 ∩ DIP* expression pattern (*Source data 5*). Knocking down *DIP-ε* in all *DIP-α* neurons changes the *fru P1 ∩ DIP-α* pattern (*Figure 10*). Males show a significant loss of neuronal projections that have 'U' shaped arbors (see *Figure 10C*, subpanel I). Both males and females show a reduction of a set of descending neurons when compared to control flies expressing *RFP RNAi* (see *Figure 10C*, subpanel II). In addition, females show an

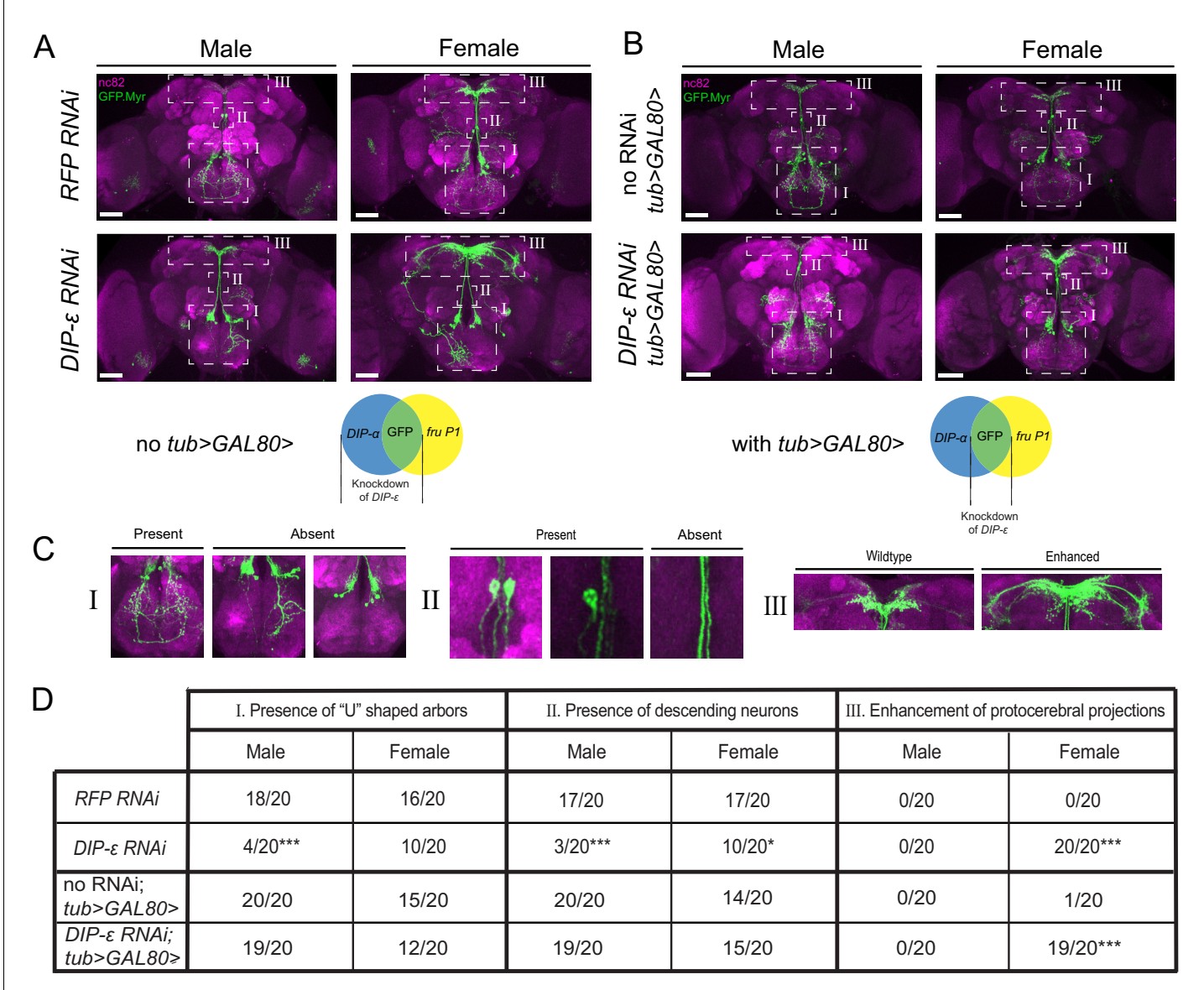

**Figure 10.** RNAi mediated knockdown of *DIP-ε* in *fru P1 ∩ DIP-α* neurons results in perturbations. Maximum intensity projections of brains of 4–7 days old adult flies showing *fru P1 ∩ DIP-α* neurons stained with anti-GFP (green; rabbit α-GFP Alexa Flour 488) and the neuropil marker nc82 (magenta; mouse α-nc82, Alexa Flour 633). (**A**) *fru P1 ∩ DIP-α* neurons with *DIP-ε* or *RFP* (control) knockdown in all *DIP-α* expressing neurons. Genotypes are *DIP-α Gal4; UAS > stop > GFP.Myr / RNAi; fruᶠᴸᴾ / +* with RNAi indicating either *RFP* or *DIP-ε* RNAi. (**B**) *fru P1 ∩ DIP-α* neurons with *DIP-ε* or no knockdown (control) restricted to only the visualized neurons (GFP+) through use of *tub>GAL80>*. Genotypes are *DIP-α Gal4; UAS > stop > GFP.Myr / RNAi; fruᶠᴸᴾ / tub>GAL80>* with RNAi indicating either *DIP-ε* or no RNAi. The neuronal populations with RNAi expression are illustrated in the Venn diagrams. White dashed boxes indicate phenotypes of interest, which are located in (**C**) and include (subpanel I) presence of the U-shaped arbors, (subpanel II) presence of at least one descending neuron, and (subpanel III) enhancement of protocerebral projections. All phenotypes were scored blind and are quantified in (**D**). Statistical significance in between control flies and *DIP-ε* RNAi flies was evaluated by the Fisher's exact test. In this figure, signicance is indicated as follows: *p<0.05, **p<0.01, ***p<0.001. n = 20 brains for each category. Magnification is ×20 and scale bars represent 50 μM.

enhancement of projections in the SMP region of the brain (see *Figure 10C*, subpanel III). These enhanced SMP projections have longer and more extensive projections that are not observed in males. Given that no other *dpr* or *DIP RNAi* perturbation shows these three phenotypes, this suggests that they are specific to the *DIP-ε* perturbation (*Source data 5*). No obvious morphological changes are observed in the ventral nerve cord.

We next examined the phenotypes when the *DIP-ε RNAi* knockdown is limited to only the *fru P1* ⋂ *DIP-α* co-expressing neurons, rather than all *DIP-α* neurons. We continue to use the genetic intersecting approach to visualize the neurons with GFP. To restrict expression of *DIP-ε RNAi* to *fru P1* ⋂ *DIP-α* neurons, we use an additional construct (*tub>GAL80>*), such that Gal4 is now only transcriptionally active in *fru P1* ⋂ *DIP-α* (*Figure 10B*). Males no longer show a significant reduction of the 'U' shaped projections, and neither sex shows a significant reduction of descending neurons (*Figure 10*). This suggests that these phenotypes are due to reduction of *DIP-ε* outside of *fru P1* ⋂ *DIP-α* neurons, in a non-cell-autonomous manner. Conversely, females still have the enhanced projections in the protocerebrum. This suggests that this phenotype is cell autonomous and driven by a reduction in *DIP-ε* expression inside the *fru P1* ⋂ *DIP-α* neurons. The results are consistent with the observation that both *fru P1* ⋂ *DIP-α* and *DIP-ε* are expressed in similar patterns in the SMP (*Figure 3*) and so it is not unexpected that expression of *DIP-ε RNAi* can have a functional impact in *fru P1* ⋂ *DIP-α*. Taken together, these results suggests that *DIP-ε* plays a critical role in establishing wildtype *fru P1* neuronal patterns, in both a cell-autonomous and non-cell-autonomous manner.

## Discussion

Based on our reevaluation of previous genomic data sets and the microscopy results presented, we show that *dprs/DIPs* are regulated by Fru[M] and expressed in *fru P1* neurons in both males and females (*Figures 1–3*). A reanalysis of five genomic studies showed that nearly all the *dprs/DIPs* are found in at least three of the *fru P1* analyses (*Goldman and Arbeitman, 2007*; *Dalton et al., 2013*; *Neville et al., 2014*; *Vernes, 2015*; *Newell et al., 2016*). It is not unexpected that there are differences across the genomic studies, given the differences in technologies (microarray and Illumina platforms), genomic approaches (RNA-seq, ChIP-seq and DAM-ID), cell types and life stages (tissue culture, pupae, adult heads, and adult *fru P1* neurons), and statistical criteria used across these studies. The expression pattern for each *fru P1* ⋂ *dpr/DIP* genotype is unique, although many genotypes have expression in the same brain regions, including the lateral protocerebral complex, mushroom body, antennal lobe, tritocerebral loop, mesothoracic triangle, and abdominal ganglion (*Figure 7*). These regions were previously shown to be among those with the most pronounced sexual dimorphism in *fru P1* neurons (*Cachero et al., 2010*; *Yu et al., 2010*). Furthermore, while the patterns for each genotype are similar between males and females, we find sexual dimorphism in some of the projection patterns and in neuron numbers (*Figures 2* and *3*). Given that the *dprs/DIPs* are not sex-specifically expressed, this suggests that their role in generating sexual dimorphism may be quantitative, due to sexual dimorphism in expression levels or differences in the number of neurons in which they are expressed in a given region.

A single-cell RNA-seq analysis of *fru P1* neurons shows that *dprs/DIPs* are expressed in every *fru P1* neuron, with the majority having a unique expression combination. Consistent with other single-cell and cell-type-specific studies of the *Drosophila* nervous system (*Cosmanescu et al., 2018*; *Croset et al., 2018*; *Davie et al., 2018*; *Allen et al., 2020*; *Davis et al., 2020*), our UMAP cluster analysis shows that *DIP*s generally have high average expression in a small set of neurons, whereas *dpr*s have moderate average expression levels in a larger set of neurons (*Figure 8*). This suggests that *dpr*s and *DIP*s may have different functional roles. Understanding how single-cell RNA-seq expression values correlate with whether the protein is produced is an important next goal, especially since the single-cell RNA-seq expression values for *dprs/DIPs* in *fru P1* neurons appears to be broader than observed in our immunofluorescence analyses presented here.

A higher resolution analysis of *fru P1* ⋂ *DIP-α* neurons found additional sexual dimorphism in projections and neuron number that are downstream of the sex hierarchy (*Figure 9*). Regulation by the sex hierarchy of *fru P1* ⋂ *DIP-α* neurons is both cell-autonomous and cell non-autonomous. These results point to the importance of understanding the development and function of *fru P1* neurons in a broad context, taking into account interactions with both *fru P1* and non-*fru P1* neurons. Furthermore, an RNAi and overexpression screen show there is functional redundancy in patterning, with

only *DIP-ε RNAi* generating phenotypes (*Figure 10*). Functional redundancy has been observed for other IgSF members (see *Schwarz et al., 2009*; *Li et al., 2017*; *Sanes and Zipursky, 2020*). In the single-cell RNA-seq analysis, *DIP-ε* had restricted expression, with limited overlap with other *dprs/DIPs* in the UMAP plot, which may point to an essential function in these cells, consistent with *DIP-ε* having an RNAi phenotype.

## Role of Dprs and DIPs in sexual dimorphism of *fru P1* neurons

In the optic lobe, antennal lobe and neuromuscular junction, genetic analyses have demonstrated that Dprs/DIPs have a role in synaptic specificity and connectivity, with Dpr-DIP interactome partner pairs mediating these critical functions (*Carrillo et al., 2015*; *Tan et al., 2015*; *Barish et al., 2018*; *Xu et al., 2018*; *Ashley et al., 2019*; *Courgeon and Desplan, 2019*; *Menon et al., 2019*; *Venkatasubramanian et al., 2019*; *Xu et al., 2019*). Our screen to identify morphological or synaptic changes in *fru P1 ∩ DIP-α* and *DIP-δ* neurons, using *dpr/DIP* RNAi or overexpression transgenes identified only one perturbation with an impact: reduction of *DIP-ε* by RNAi on the *fru P1 ∩ DIP-α* pattern, with both cell autonomous and non-autonomous roles (*Figure 10*). This suggests that there is sufficient redundancy in these neurons that removal or addition of one member of the Dpr/DIP interactome cannot change patterning robustly. DIP-ε interacts with a large number of Dprs, which may be one of the reasons a reduction of DIP-ε results in morphological changes. We found that some Dprs that interact with the same DIP are expressed in the same brain regions (*Figure 7*), and/or are detected in the same neurons, consistent with the idea of redundancy. This could also be due to other members of the IgSF that were identified by our genomic-scale screens as expressed in *fru P1* neurons and/or are regulated by Fru$^M$, or other adhesion molecules. The enhanced set of projections in the superior medial protocerebrum region of the brain due to reduced *DIP-ε* is reminiscent of the synaptic targeting phenotypes seen in the optic lobe due to *dpr/DIP* perturbations (*Carrillo et al., 2015*; *Tan et al., 2015*; *Courgeon and Desplan, 2019*; *Menon et al., 2019*; *Xu et al., 2019*), which supports a role of *dprs/DIPs* in the development of *fru P1* neuroanatomical projection patterns and/or synaptic targets.

Future studies that are performed with genetic tools that yield more penetrant phenotypes than RNAi, including CRISPR/Cas9 generated alleles, will likely reveal additional roles for Dprs/DIPs and can provide further confirmation of the RNAi results presented. Furthermore, CRISPR/Cas9 gene knock-out approaches can target multiple *dprs/DIPs*, allowing one to test for functional redundancy. Additional analyses to determine the subcellular localization of each Dpr/DIP will also be important to understand their roles in the nervous system, especially to determine if they are present in synaptic termini and dendrites, which would be consistent with a role of directing synaptic specificity. It is clear that higher resolution analyses of the *fru P1 ∩ DIP-α* pattern revealed more sexual dimorphism, so additional analysis of other genotypes at this resolution will be important, including determining developmental patterns to gain insight into mechanisms that underlie sexual dimorphism. Furthermore, our expression data reveal expression beyond development, well into adult stages. Adult roles of the *dprs/DIPs* may include mediating neuronal connectivity changes due to reproductive experiences.

## *fru P1 ∩ dpr/DIP* neurons and male courtship behaviors

We find that activating and silencing the subsets of neurons defined by each *fru P1 ∩ dpr/DIP* genotype differentially impacts male courtship behaviors, with the results highlighting that the activity of different combinations of neurons can generate a similar behavioral outcome (*Figure 7*). This analysis provides further insight into how similar behavioral outcomes can be generated in different ethological contexts, through the integration of information across many different neuronal subtypes. An examination of the similarities of *fru P1 ∩ dpr/DIP* expression patterns and behavioral outcomes suggests that interactions between the mushroom body and lateral protocerebral complex are critical to reach a certain threshold of activation for male courtship behaviors, in both the male-female and male-alone paradigm, given that the genotypes with expression in those two regions had the most consistent and robust behavioral phenotypes. Interactions between neurons in these two regions have previously been proposed to integrate disparate sensory information and behavioral experiences in order to direct courtship outcomes (*Yu et al., 2010*).

While there has been an impressive effort to map functions onto small subsets of neurons (*Robie et al., 2017*), our results suggest that it will also be important to understand the roles of different combinations of neurons to fully understand behavioral outcomes. This will facilitate understanding of how different sensory and courtship experiences impart physiological changes to direct behavior. Furthermore, these activation experiments may also reveal insights about evolution of behavior. In some *Drosophila* species, males perform a double wing extension during courtship (reviewed in *Anholt et al., 2020*). We observe double wing extension in several genotypes in the neuronal activation experiments, suggesting that changing levels of neuronal activity are a way to evolve a new behavior. Although this study focused on male reproductive behaviors, it will also be interesting to examine the role of *fru P1 ∩ dpr/DIP* neurons on female behavioral outcomes.

## Conclusions

Over the last several years, genomic studies have pointed to a role of the *dprs/DIPs* in *fru P1* neurons (*Goldman and Arbeitman, 2007*; *Dalton et al., 2013*; *Neville et al., 2014*; *Vernes, 2015*; *Newell et al., 2016*). Indeed, our early study showed that *dpr1* had a role in courtship gating, or the timing of the steps that the male performs (*Goldman and Arbeitman, 2007*). Until recently, a systematic analysis of the role of *fru P1* neurons that express *dprs/DIPs* was not possible. The use of the intersecting genetic strategy with the set of *dpr/DIP Gal4* tools revealed their expression patterns and that there is an interplay between mushroom body and lateral protocerebral complex neurons, with respect to coordinating behavioral outcomes. In addition, focused experiments on the *fru P1 ∩ DIP-α* pattern revealed roles for the sex hierarchy and the functional redundancy of *dprs/DIPs*, based on the RNAi perturbation results. Future functional studies aimed at a systematic analysis of the Dpr/DIP interactome will further elucidate the role of these cell adhesion molecules in terms of specifying neuroanatomy and also as a powerful tool to gain insight into the functions of different sets of *fru P1* neurons.

# Materials and methods

**Key resources table**

| Reagent type (species) or resource | Designation | Source or reference | Identifiers | Additional information |
|---|---|---|---|---|
| Genetic reagent (*D. melanogaster*) | dpr1-Gal4 | Provided by Zinn lab | | w;dpr1-T2A-GAL4/cyo |
| Genetic reagent (*D. melanogaster*) | dpr2-Gal4 | Provided by Zinn lab | | w;dpr2-T2A-GAL4/cyo |
| Genetic reagent (*D. melanogaster*) | dpr3-Gal4 | Provided by Zinn lab | | w;dpr3-T2A-GAL4/cyo (#2) |
| Genetic reagent (*D. melanogaster*) | dpr4-Gal4 Flybase symbol: Dmel\TI{CRIMIC.TG4.1}dpr4[CR00485-TG4.1] | Hugo Bellen/ Gene Disruption Project (GDP) *Lee et al., 2018* PMID:29565247 | RRID:BDSC_79271 | y(1) w[*]; TI{GFP[3xP3.cLa]= CRIMIC.TG4.1}dpr4[CR00485-TG4.1]/TM3, Sb(1) Ser(1) |
| Genetic reagent (*D. melanogaster*) | dpr5-Gal4 | Provided by Zinn lab | | w;;dpr5-T2A-GAL4/TM6b (#2) |
| Genetic reagent (*D. melanogaster*) | dpr6-Gal4 | Provided by Zipursky *Tan et al., 2015* PMID:26687360 | | dpr6-T2A-GAL4/TM6b |
| Genetic reagent (*D. melanogaster*) | dpr7-Gal4 Flybase symbol: Dmel\Mi{MIC}dpr7[MI05719] | Hugo Bellen/ Gene Disruption Project (GDP) *Nagarkar-Jaiswal et al., 2015a* PMID:25824290 | RRID:BDSC_60786 | Y(1)Mi{y[+Dint2]=MIC} dpr7[MI05719] |
| Genetic reagent (*D. melanogaster*) | dpr8-Gal4 | Provided by Zinn lab | | dpr8-GAL4/Fm7a |
| Genetic reagent (*D. melanogaster*) | dpr9-Gal4 | Provided by Zinn lab | | w; BL/Cyo; dpr9-T2A-GAL4/TM6b |

*Continued on next page*

*Continued*

| Reagent type (species) or resource | Designation | Source or reference | Identifiers | Additional information |
|---|---|---|---|---|
| Genetic reagent (*D. melanogaster*) | dpr10-Gal4 | Provided by Zinn lab | | *w;;dpr10-T2A-GAL4/TM6b (#3)* |
| Genetic reagent (*D. melanogaster*) | dpr11-Gal4 | Provided by Zinn lab | | *w; BL/Cyo; dpr11-GAL4/TM6b* |
| Genetic reagent (*D. melanogaster*) | dpr12-Gal4 | Provided by Zinn lab | | *w; dpr12-GAL4/cyo; TM2/TM6b* |
| Genetic reagent (*D. melanogaster*) | dpr13-Gal4 | Provided by Zinn lab | | *w;dpr13-GAL4/cyo; TM2/TM6b* |
| Genetic reagent (*D. melanogaster*) | dpr14-Gal4 Flybase symbol: Dmel\TI{CRIMIC.TG4.1} dpr14[CR00516-TG4.1] | Hugo Bellen/ Gene Disruption Project (GDP) *Lee et al., 2018* PMID:29565247 | RRID:BDSC_80586 | *y(1) TI{GFP[3xP3.cLa]= CRIMIC.TG4.1}dpr14 [CR00516-TG4.1] w[*]* |
| Genetic reagent (*D. melanogaster*) | dpr15-Gal4 Flybase symbol: Dmel\Mi{Trojan-GAL4.1} dpr15[MI01408-TG4.1] | Hugo Bellen/ Gene Disruption Project (GDP) *Lee et al., 2018* PMID:29565247 | RRID:BDSC_66827 | *y(1) w[*];; Mi{Trojan-GAL4.1}dpr15 [MI01408-TG4.1]/TM3, Sb(1) Ser(1)* |
| Genetic reagent (*D. melanogaster*) | dpr16-Gal4 | Provided by Zinn lab | | *w;;dpr16-T2A-GAL4/TM6b (#3)* |
| Genetic reagent (*D. melanogaster*) | dpr17-Gal4 Flybase symbol: Dmel\Mi{Trojan-GAL4.1} dpr17[MI08707-TG4.1] | Hugo Bellen/ Gene Disruption Project (GDP) *Lee et al., 2018* PMID:29565247 | RRID:BDSC_76200 | *y(1) w[*]; Mi{Trojan-GAL4.1} dpr17[MI08707-TG4.1]/TM3, Sb(1) Ser(1)* |
| Genetic reagent (*D. melanogaster*) | dpr18-Gal4 Flybase symbol: Dmel\TI{CRIMIC.TG4.1} dpr18[CR01004-TG4.1] | Hugo Bellen/ Gene Disruption Project (GDP) *Lee et al., 2018* PMID:29565247 | RRID:BDSC_83245 | *y(1) w[*]; TI{GFP[3xP3.cLa]=CRIMIC.TG4.1} dpr18[CR01004-TG4.1]* |
| Genetic reagent (*D. melanogaster*) | dpr19-Gal4 Flybase symbol: Dmel\TI{CRIMIC.TG4.1} dpr19[CR00996-TG4.1] | Hugo Bellen/ Gene Disruption Project (GDP) *Lee et al., 2018* PMID:29565247 | RRID:BDSC_83242 | *y(1) w[*]; TI{GFP[3xP3.cLa]=CRIMIC.TG4.1} dpr19[CR00996-TG4.1]/SM6a* |
| Genetic reagent (*D. melanogaster*) | DIP-$\alpha$-Gal4 | Provided by Zinn lab | | *DIP-alpha-T2A-GAL4/(Fm7)* |
| Genetic reagent (*D. melanogaster*) | DIP-$\beta$-Gal4 | Provided by Zinn lab | | DIP-beta--T2A-GAL4GAL4 |
| Genetic reagent (*D. melanogaster*) | DIP-$\gamma$-Gal4 | Provided by Zinn lab | | *w;;DIP-gamma--T2A-GAL4GAL4/TM3, Sb (#8)* |
| Genetic reagent (*D. melanogaster*) | DIP-$\delta$-Gal4 | Provided by Zinn lab | | *w;;DIP-delta--T2A-GAL4GAL4/TM6b (#3)* |
| Genetic reagent (*D. melanogaster*) | DIP-$\epsilon$-Gal4 Flybase symbol: Dmel\Mi{Trojan-GAL4.1} DIP-$\epsilon$[MI11827-TG4.1] | Hugo Bellen/ Gene Disruption Project (GDP) *Lee et al., 2018* PMID:29565247 | RRID:BDSC_67502 | |
| Genetic reagent (*D. melanogaster*) | DIP-$\zeta$-Gal4 | Provided by Zinn lab | | *w;DIP-zeta-T2A-GAL4/Cyo* |
| Genetic reagent (*D. melanogaster*) | DIP-$\eta$-Gal4 | Provided by Zinn lab | | *w;DIP-eta-T2A-GAL4/Cyo* |
| Genetic reagent (*D. melanogaster*) | DIP-$\theta$-Gal4 | Provided by Zinn lab | | *w;DIP-theta-Gal4/cyo;TM2/TM6b* |
| Genetic reagent (*D. melanogaster*) | DIP-$\iota$-Gal4 Flybase symbol: Dmel\TI{CRIMIC.TG4.1} DIP-$\iota$[CR00997-TG4.1] | Hugo Bellen/ Gene Disruption Project (GDP) *Lee et al., 2018* PMID:29565247 | RRID:BDSC_83243 | *y(1) w[*]; TI{GFP[3xP3.cLa]=CRIMIC.TG4.1} DIP-iota[CR00997-TG4.1]/SM6a* |
| Genetic reagent (*D. melanogaster*) | 10xUAS > stop > myr::smGdP-cMyc Flybase symbol: Dmel\P{10XUAS(FRT.stop) GFP.Myr}su(Hw)attP5 | Pfeiffer, B., Rubin, G. (2014.4.16). Recombinase and tester constructs and insertions. Flybase ID: FBrf0224689 | RRID:BDSC_55810 | *+/+; P{10XUAS(FRT.stop)GFP.Myr} su(Hw)attP5; +/+* |

*Continued on next page*

*Continued*

| Reagent type (species) or resource | Designation | Source or reference | Identifiers | Additional information |
|---|---|---|---|---|
| Genetic reagent (*D. melanogaster*) | Canton S | Ulrike Heberlein | | Wild type |
| Genetic reagent (*D. melanogaster*) | Canton S (white) | Ulrike Heberlein | | Wild type with *white* mutation introgression |
| Genetic reagent (*D. melanogaster*) | *UAS > stop > TrpA1* Flybase symbol: Dmel\P{UAS(FRT.stop) TrpA1[myc]}VIE-260B | Dickson, B. (2017.1.31). Barry Dickson Stocks. Flybase ID: FBrf0234603 | RRID:BDSC_66871 | w[*]; P{y[+t*] w[+mC]= UAS(FRT.stop) TrpA1[myc]}VIE-260B |
| Genetic reagent (*D. melanogaster*) | *fru P1*[FLP] Flybase symbol: Dmel\TI{FLP}fru[FLP] | Donor: Barry Dickson, Howard Hughes Medical Institute, Janelia Research Campus | RRID:BDSC_66870 | w[*]; TI{FLP}fru[FLP]/TM3, Sb(1) |
| Genetic reagent (*D. melanogaster*) | *UAS > stop > TNTQ* Flybase symbol: P{UAS(FRT.stop)Ctet \tetX[in]} | Barry Dickson *Stockinger et al., 2005* PMID:15935765 | | w[*]; P{w[+m*]=UAS(FRT.stop) Ctet\tetX[in]}VIE-19A/CyO |
| Genetic reagent (*D. melanogaster*) | *UAS > stop > TNTE* Flybase symbol: Dmel\P{UAS(FRT.stop) Ctet\tetX}VIE-19A | Barry Dickson, Howard Hughes Medical Institute, Janelia Research Campus | RRID:BDSC_67690 | w[*]; P{w[+m*]=UAS(FRT.stop) Ctet\tetX}VIE-19A/CyO |
| Genetic reagent (*D. melanogaster*) | *tub >GAL80>* Flybase symbol: Dmel\P{αTub84B (FRT.GAL80)}3 | Zhang, B. (2012.5.29). P{alphaTub84B(FRT.GAL80)} insertions from Bing Zhang Flybase ID: FBrf0218396 | RRID:BDSC_38881 | w[*]; Bl(1)/CyO; P{w[+mC]= alphaTub84B(FRT.GAL80)}3 |
| Genetic reagent (*D. melanogaster*) | *UAS-DIP-α* | Provided by Zipursky *Xu et al., 2018* PMID:30467079 | | w;BL/CyO;UAS-Dipalpha-2ATdTom/TM6b |
| Genetic reagent (*D. melanogaster*) | *RFP-RNAi* Flybase symbol: Dmel\P{TRiP.HMS05847} attP40 | Donor: Transgenic RNAi Project *Perkins et al., 2015* PMID:26320097 | RRID:BDSC_67984 | y(1) v(1); P{y[+t7.7] v[+t1.8]= TRiP.HMS05847}attP40 |
| Genetic reagent (*D. melanogaster*) | *DIP-α-RNAi* Flybase symbol: Dmel\P{TRiP.HMS05847} attP40 | Donor: Transgenic RNAi Project *Perkins et al., 2015* PMID:26320097 | RRID:BDSC_67984 | y(1) v(1); P{y[+t7.7] v[+t1.8]= TRiP.HMS01879}attP40/CyO |
| Genetic reagent (*D. melanogaster*) | *DIP-β RNAi* Flybase symbol: Dmel\P{TRiP.HMS01774} attP40 | Donor: Transgenic RNAi Project *Perkins et al., 2015* PMID:26320097 | RRID:BDSC_38310 | y(1) sc[*] v(1); P{y[+t7.7] v[+t1.8]= TRiP.HMS01774}attP40 |
| Genetic reagent (*D. melanogaster*) | *DIP-ε-RNAi* Flybase symbol: Dmel\P{TRiP.HMS01718} attP40 | Donor: Transgenic RNAi Project *Perkins et al., 2015* PMID:26320097 | RRID:BDSC_38936 | y(1) sc[*] v(1); P{y[+t7.7] v[+t1.8]= TRiP.HMS01718}attP40 |
| Genetic reagent (*D. melanogaster*) | *DIP-η-RNAi* Flybase symbol: Dmel\P{TRiP.HMS01673} attP40 | Donor: Transgenic RNAi Project *Perkins et al., 2015* PMID:26320097 | RRID:BDSC_38229 | y(1) sc[*] v(1); P{y[+t7.7] v[+t1.8]= TRiP.HMS01673}attP40 |
| Genetic reagent (*D. melanogaster*) | *DIP-ι-RNAi* Flybase symbol: Dmel\P{TRiP.HMS01675} attP40 | Donor: Transgenic RNAi Project *Perkins et al., 2015* PMID:26320097 | RRID:BDSC_38231 | y(1) sc[*] v(1); P{y[+t7.7] v[+t1.8]= TRiP.HMS01675}attP40 |
| Genetic reagent (*D. melanogaster*) | *DIP-θ-RNAi* Flybase symbol: Dmel\P{TRiP.JF03069} attP2 | Donor: Transgenic RNAi Project *Perkins et al., 2015* PMID:26320097 | RRID:BDSC_28654 | y(1) v(1);; P{y[+t7.7] v[+t1.8]= TRiP.JF03069}attP2 |

*Continued on next page*

*Continued*

| Reagent type (species) or resource | Designation | Source or reference | Identifiers | Additional information |
|---|---|---|---|---|
| Genetic reagent (*D. melanogaster*) | *DIP-ζ-RNAi* Flybase symbol: Dmel\P{TRiP.HMS01671} attP40 | Donor: Transgenic RNAi Project *Perkins et al., 2015* PMID:26320097 | RRID:BDSC_38227 | *y(1) sc[*] v(1); P{y[+t7.7] v[+t1.8]= TRiP.HMS01671}attP40* |
| Genetic reagent (*D. melanogaster*) | *dpr5-RNAi* Flybase symbol: Dmel\P{TRiP.JF03306} attP2 | Donor: Transgenic RNAi Project *Perkins et al., 2015* PMID:26320097 | RRID:BDSC_29627 | *y(1) v(1);; P{y[+t7.7] v[+t1.8]= TRiP.JF03306}attP2* |
| Genetic reagent (*D. melanogaster*) | *dpr10-RNAi* Flybase symbol: Dmel\P{TRiP.JF02920} attP2 | Donor: Transgenic RNAi Project *Perkins et al., 2015* PMID:26320097 | RRID:BDSC_27991 | *y(1) v(1);; P{y[+t7.7] v[+t1.8]= TRiP.JF02920}attP2* |
| Genetic reagent (*D. melanogaster*) | *dpr12-RNAi* Flybase symbol: Dmel\P{TRiP.JF03210} attP2 | Donor: Transgenic RNAi Project *Perkins et al., 2015* PMID:26320097 | RRID:BDSC_28782 | *y(1) v(1);; P{y[+t7.7] v[+t1.8]= TRiP.JF03210}attP2* |
| Genetic reagent (*D. melanogaster*) | UAS-DIP-ε | FlyORF PMID:23637332 and 24922270 | Fly line ID: F004486 | ;;M{UAS-DIP-epsilon.ORF.3xHA.GW}ZH-86Fb* |
| Genetic reagent (*D. melanogaster*) | UAS-DIP-γ | FlyORF PMID:23637332 and 24922270 | Fly line ID: F003086 | ;;M{UAS-DIP-gamma.ORF.3xHA.GW}ZH-86Fb* |
| Genetic reagent (*D. melanogaster*) | UAS-DIP-ι | FlyORF PMID:23637332 and 24922270 | Fly line ID: F004254 | ;;M{UAS-DIP-iota.ORF.3xHA.GW}ZH-86Fb* |
| Genetic reagent (*D. melanogaster*) | UAS-dpr1 | FlyORF PMID:23637332 and 24922270 | Fly line ID: F004145 | ;;M{UAS-dpr1.ORF.3xHA.GW}ZH-86Fb |
| Genetic reagent (*D. melanogaster*) | UAS-dpr4 | FlyORF PMID:23637332 and 24922270 | Fly line ID: F002762 | ;;M{UAS-dpr4.ORF. 3xHA.GW}ZH-86Fb |
| Genetic reagent (*D. melanogaster*) | UAS-dpr6 | Provided by Zipursky *Xu et al., 2018* PMID:30467079 | | w;UAS-Dpr6F-V5/CyO;TM2/TM6b |
| Genetic reagent (*D. melanogaster*) | UAS-dpr10 | Provided by Zipursky *Xu et al., 2018* PMID:30467079 | | w;UAS-Dpr10D-V5/CyO;TM2/TM6b |
| Genetic reagent (*D. melanogaster*) | UAS-dpr18 | FlyORF PMID:23637332 and 24922270 | | ;;M{UAS-dpr18.ORF.3xHA.GW}ZH-86Fb* |
| Genetic reagent (*D. melanogaster*) | *UAS-Tra^F* Flybase symbol: Dmel\P{UAS-tra.F}20J7 | Donor: Ralph Greenspan, New York University | RRID:BDSC_4590 | w[1118]; P{w[+mC]=UAS-tra.F}20J7 |
| Genetic reagent (*D. melanogaster*) | *UAS-Fru^MA7* | Provided by Stephen Goodwin | | |
| Genetic reagent (*D. melanogaster*) | *UAS-Fru^MB25* | Provided by Stephen Goodwin | | |
| Genetic reagent (*D. melanogaster*) | *UAS-Fru^MC1* | Provided by Stephen Goodwin | | |
| Genetic reagent (*D. melanogaster*) | *Fru^AC* | PMID:16753560 | | |
| Genetic reagent (*D. melanogaster*) | Flybase symbol: Dmel\P{10XUAS-IVS-mCD8::GFP}attP40 | Donor: Gerald M. Rubin and Barret Pfeiffer, Howard Hughes Medical Institute, Janelia Research Campus | RRID:BDSC_32186 | w[*]; P{y[+t7.7] w[+mC]=10XUAS -IVS-mCD8::GFP }attP40 |
| Genetic reagent (*D. melanogaster*) | *fru P1-Gal4* | Provided by Baker lab PMID:15959468 | | |

## Fly husbandry and stocks

All flies were raised at 25°C on a 12:12 hr light-dark cycle. The flies were grown using standard corn-meal food media (33 L $H_2O$, 237 g Agar, 825 g dried deactivated yeast, 1560 g cornmeal, 3300 g dextrose, 52.5 g Tegosept in 270 ml 95% ethanol and 60 ml Propionic acid).

A list of *Drosophila* strains is provided (Key Resources Table).

## Immunohistochemistry and confocal microscopy

Brain and ventral nerve cord (VNC) tissues were dissected from animals that were either 0–24 hr adults, or 4–7 day adults. Samples were dissected in 1x Phosphate Buffered Saline (PBS; 140 mM NaCl, 10 mM phosphate buffer, and 3 mM KCl, pH 7.4) and immediately transferred to fix (4% para-formaldehyde, 1x PBS) for 25 min at room temperature. Samples were washed for 5 min with 1x PBS, three times. The tissue was then permeabilized with TNT (0.1 M Tris-HCl [pH 7.4], 0.3 M NaCl, 0.5% Triton X-100), for 15 min, followed by two additional 5 min TNT washes. The tissue was rinsed in 1x PBS, and then Image-iT FX Signal Enhancer (Invitrogen) was applied for 25 min. Finally, the tissue was washed in TNT for two washes of 5 min each. Diluted primary antibody in TNT was applied, and samples were incubated overnight at 4°C. Next, the tissue was washed six times in TNT for 5 min each, and then secondary antibody diluted in TNT and applied. The samples were then incubated for 2 hr at room temperature or overnight at 4°C. Following this incubation, samples were washed six times in TNT for 5 min each and then mounted in Secureseal Image Spacers (Electron Microscopy Services), on glass slides with VectaShield Mounting Medium (Vector Laboratories; H-1000), and covered with #1.5 coverslips. Primary antibodies were used in the following dilutions, as indicated in the figure legends: mouse α-nc82 (1:20; Developmental Studies Hybridoma Bank, AB_2314866), rabbit α-Myc (1:6050; abcam, ab9106), rabbit α-GFP Alexa Fluor 488 (1:600; Invitrogen, A21311). Secondary antibodies were used in the following dilutions: goat α-rabbit Alexa Fluor 568 (1:500; Invitrogen, A11036), goat α-mouse Alexa Fluor 633 (1:500; Invitrogen A21052). For labeling of three MCFO markers (FLAG, V5, and HA), brains and VNCs samples were dissected and stained by following the method modified from *Nern et al., 2015*. The primary antibodies rabbit α-HA (1:300; Cell Signaling, 3724S), mouse α-FLAG (1:500; Sigma, F1804), and rabbit α-V5 DyLight 549 (Rockland, 600-442-378), and the secondary antibodies goat α-rabbit Alexa Fluor 633 (1:500; Invitrogen A21071) and goat α-mouse Alexa Fluor 488 (1:500; Invitrogen A11001) were used. All the antibodies were diluted in TNT.

Images were acquired on a Zeiss LSM 700 confocal microscope with a 20x objective and bidirectional scanning. The interval of each slice was set as 1.0 μm. Zeiss Zen software (Black edition, 2012) was used to make adjustments to laser power and detector gain to enhance the signal to noise ratio.

## Image analysis and quantification of *fru P1* ∩ *dpr/DIP* neurons

Brain and VNC confocal images of 4- to 7-day-old male or female adults were analyzed for the presence of certain morphological features and cell body numbers of select neurons. The images were scored blind, in randomized batches, by three independent people. The analysis was performed using Fiji-ImageJ 14.1, with the cell counter Janelia version 1.47 hr plugin. To determine which regions to analyze, the following criteria were used: (1) regions that had sexually dimorphic structures, (2) were present in many of the different genotypes, and/or (3) are known to be important for reproductive behaviors. A template of example images, with regions indicated, was used to ensure accurate and similar image analyses across all researchers (*Source data 1*). As a test to ensure accuracy of scoring across the three individuals, a round-robin scoring design was employed, with each image scored by three individuals, for a subset of 26 images, which showed high concordance. The raw cell count numbers and morphological observations were recorded in excel, compiled and then unblinded (*Source data 1*).

## Generation of heatmaps

Heatmaps and correlation plots of the image analysis data were generated using Morpheus (Broad Institute; https://software.broadinstitute.org/morpheus). For features that were scored as present or absent, a value of 0 or one was calculated as the number of samples with the feature present divided by total number of samples. For the cell count data, the replicate data was averaged, and then all data was divided by the highest value for that cell count feature, so all data were between 0 and 1. The hierarchal cluster heatmap was made using the following parameters: one minus spearman rank correlation as the metric, average for linkage method, and clustering by the columns (data for each

*dpr/DIP*). The correlation heatmaps were created using the Morpheus similarity matrix tools, using the following parameters: spearman rank as the metric, computed for the columns.

## Courtship behavior assays and analyses

For all behavior, male flies were collected 0–6 hr post-eclosion, housed individually in small vials, and aged for 4–7 days. Canton S virgin females (*white*) were also collected 0–6 hr post-eclosion, and aged for 4–7 days in groups to be used as female targets for courtship with males containing the *UAS > stop > TrpA1:myc* transgene. Canton S virgin females were collected and kept in a similar manner to be used for courtship with male flies containing the *UAS > stop > TNT/TNTQ* transgenes. Flies were kept in a 25°C incubator on a 12:12 hr light:dark cycle, unless otherwise noted. Courtship chambers were placed on a temperature-controlled metal block at 25°C and videos were recorded between ZT 5–10, in a 10 mm chamber for 10 min, or until successful copulation occurred, whichever came first. For courtship using male flies harboring the *UAS > stop > TrpA1:myc* transgene, the male flies were reared and housed in a 19°C incubator, on a 12:12 light:dark cycle, so the Trp channel would not be activated. Courtship chambers were placed on a temperature-controlled metal block for ten minutes prior to the courtship assay, at either 20°C or 32°C.

The courtship video recordings were analyzed using The Observer XT (Noldus) (version 14.0), with an n = 14–16 for male-female behavior and n = 10 for male alone behavior. Coded behaviors included: following (a start-stop event defined as any time the male is oriented toward the female and is less than half a chamber distance away from the female), wing extension (a start-stop event defined as any time one wing is extended from the fly and is vibrating), double wing extension (a start-stop event when both wings are extended from the body and are vibrating), abdominal bending (a start-stop event when the abdomen is curled under and is not thrusting or is not in the correct position to copulate with the female), motor defect (a start-stop event when the male falls onto his back and is unable to right himself), attempted copulation (a point event when the male attempts to copulate with the female but is not successful), and successful copulation (a point event when the male is able to attach and successfully copulate with the female).

These data were graphed and analyzed using the JMP Pro 14.0.0 statistical software. A non-parametric Wilcoxon test was used to compare differences between the control and experimental temperature (for TrpA1 experiments) or between control and experimental strains (for TNT experiments), for the data for which an index is calculated. The unpaired *t-test* was used to determine significant differences between experimental and control conditions, with the same *dpr/DIP-Gal4*, to determine if the number of attempted copulations were different (test assumes equal variance).

## *Drosophila* activity monitor behavioral assay

Males were collected 0–6 hr post-eclosion and aged for 3 days in a 25°C incubator on a 12:12 hr light:dark cycle. On day three, they were individually loaded into 5 × 65 mm glass tubes (Trikinetics Inc), plugged on one end with standard cornmeal food media dipped in paraffin wax to seal. The non-food end was sealed with parafilm, with small air holes. The vials were loaded into *Drosophila* activity monitors (TriKinetics Inc), and placed in a 25°C incubator in 12:12 hr light:dark. Each condition was run for five days. The data from the first day of activity was not used in the analysis, as flies were recovering from $CO_2$ anesthesia. Activity was measured as the number of beam breaks and collected in 5-min bins. Beam crossings were summed over the 24 hr period from day 5 ZT0 (lights-on) to day 6 ZT0 per individual fly (*Source data 2*). These data were graphed and analyzed using the JMP Pro 14.0.0 statistical software. A non-parametric Wilcoxon test was used to compare differences between the control (TNTQ) and experimental (TNT) strains with the same *dpr/DIP-Gal4*.

## Dissociation of CNS for single-cell mRNA sequencing analyses

Twenty freshly dissected male brains and ventral nerve cords, from 48 hr after puparium formation (APF) stage, were used. The flies had expression of membrane-bound GFP in *fru P1* neurons and were the following genotype: w[*]; P{y[+t7.7] w[+mC]=10XUAS-IVS-mCD8::GFP}attP40/*UAS-Gal4*; *fru P1-Gal4/+*. The tissue was dissected in cold Schneider 2 *Drosophila* culture medium (S2 medium, Gibco) and transferred to a LoBind tube (Eppendorf), containing 200 µl of S2 medium. The tissue was centrifuged at 500 g for 5 min, and then was washed with 300 µl of EBSS (Earle's Balanced Salt Solution), and centrifuged again at 500 g for 5 min. After centrifugation, the supernatant was

replaced with 100 µl of papain for disassociation (50 units/ml, Worthington) diluted in EBSS. Brains were dissociated at 25℃ in a LoBind tube for 30 min., with pipette mixing to reinforce dissociation every 3 min with a P200 tip during the first 15 min, and a P10 tip for the final 15 min. Cells were washed twice with 700 µl cold S2 medium containing 10% FBS (Gibco) and centrifuged at 700 g for 10 min to quench the papain. Cell suspensions were passed through a 30 µM pre-separation filter (Miltenyi Biotech). Cell viability and concentration were assessed by hemocytometer using Trypan blue.

## 10x Genomics library preparation and sequencing

Single-cell libraries were generated using Single Cell 3′ Library and Gel Bead Kit v2, Chip Kit, and the GemCode 10X Chromium instrument (10X Genomics, CA), according to the manufacturer's protocol (*Zheng et al., 2017*). In brief, single cells were suspended in S2 medium with 10% FBS and the maximum volume of cells, 34 µl, was added to a single chip channel. After the generation of nanoliter-scale <u>Ge</u>l bead-in-<u>EM</u>ulsions (GEMs), the mRNA in GEMs underwent reverse transcription. Next, GEMs were broken, and the single-stranded cDNA was isolated, cleaned with Cleanup Mix containing DynaBeads MyOne Silane beads (Thermo Fisher Scientific). cDNA was then amplified with the following PCR machine settings: 98℃ for 3 min, 9 cycles of (98℃ for 15 s, 67℃ for 20 s), 72℃ for 1 min, held at 4℃. Subsequently, the amplified cDNA was cleaned up with SPRIslect Reagent kit (Beckman Coulter), fragmented, end-repaired, A-tailed, adaptor ligated, and cleaned with SPRIselect magnetic beads between steps. This product was PCR amplified with the following PCR machine settings: 98℃ for 45 s, 12 cycles of (98℃ for 20 s, 54℃ for 30 s, 72℃ for 20 s), 72℃ for 1 min, and hold at 4℃. The library was cleaned and size-selected with SPRIselect beads, followed by Pippin size selection for a 350–450 bp library size range. Single cell libraries were sequenced on the Illumina NovaSeq with 150 bp paired-end reads on an S2 flowcell. This produced 1,870,220,065 reads. The sequence data are available through the GEO repository (GSE162098).

## Single-cell data pre-processing and analysis

Raw reads were processed using the CellRanger software pipeline (v.2.1.1) 'cellranger count' command to align reads to the *Drosophila melanogaster* (BDGP6.92) STAR reference genome, customized to contain the sequence for the *mCD8-GFP* cDNA. The 'force-cells' command was used to call 25,000 single cells, based on the inflection point of the CellRanger barcode rank plot, a criterion for dividing single cells from empty GEM droplets (*Source data 3*). The recovered 25,000 single cells had a mean sequencing depth of 74,808 reads per cell. We detected a median of 2118 genes per cell. The obtained feature-barcode matrix was further processed and analyzed in the R package Seurat (v3.0) (*Stuart et al., 2019*). To filter the expression matrix for high quality cells we removed cells with >5% mitochondrial transcripts (dying cells),<200 genes (empty droplets), and/or expressing more than 6000 genes (potential doublets or triplets). This filtering produced a matrix of 24,902 high-quality cells which were computationally subset to the population of *fru P1* neurons, based on *mCD8-GFP* expression, obtaining 5,621 cells. We next followed the Seurat 'Guided clustering tutorial' for default normalization and scaling steps (https://satijalab.org/seurat/v3.0/pbmc3k_tutorial.html). Expression was normalized using the 'NormalizeData' function where gene counts within each cell are divided by the total gene counts for that cell, multiplied by a scaling factor of 10000, and natural-log transformed (log1p). A linear transformation was applied to the normalized gene counts, to make genes more comparable to one another, using the default 'ScaleData' function to center the mean expression to 0 and set the variance at 1. We performed a principal component analysis (PCA) using only the data from 33 *dpr/DIP* genes. We used the top 20 principal components based on visual inspection of DimHeatmap outputs and the ElbowPlot. Selecting more than 20 PCs did not dramatically change our results. We then continued to follow Seurat's standard workflow to reduce dimensionality and cluster cells using the default 'FindNeighbors', 'FindClusters', and 'runUMAP' functions (resolution = 1.3).

To evaluate expression combinations of the *dpr/DIP*s within our single cells we used an UpSet plot analysis (*Conway et al., 2017*). To do this, we transposed our matrix which contained normalized, log-transformed, and scaled expression data (*Source data 3*) for *dpr/DIP*s for each single cell barcode and binarized the data (any expression of a *dpr* or *DIP* >1 = 1, and >1 is considered as no expression = 0, *Source data 3*). All plots generated are ordered by the highest frequency of an

expression combination occurring within single cells (order.by = 'freq'). A single-cell expression hierarchical clustering dendrogram was produced using the normalized, log-transformed, and scaled expression data (*Source data 3*). A Pearson correlation distance measure was calculated using the factoextra (v. 1.0.7) 'get_dist' function and hierarchical cluster analysis was performed using the 'hclust' core R statistics function with the argument method = 'average'.

## Image analyses of RNAi and over-expression perturbations

### Sex hierarchy perturbations

The *DIP-α* subset of *fru P1* neurons were analyzed to determine the impact of sex hierarchy perturbations. Flies bearing RNAi and over-expressor constructs were raised to 4- to 7-day-old adults, stained, and imaged as described above. Both RNAi knockdown and overexpression experiments were first performed in all *DIP-α* expressing cells. In addition, the over-expressors were also restricted to the visualized *fru P1 ∩ DIP-α* neurons with the use of *tub>GAL80>*.

The *fru P1 ∩ DIP-α* neuronal patterns were analyzed blind in at least 15 brains and ventral nerve cords, in males and females, to determine the effect of sex hierarchy perturbations on neuronal morphology (*Source data 4*), for a set of morphological features (*Source data 4*). The ratios of different types of the morphological features and presence or absence of morphological features were compared within sex, between sex hierarchy perturbation groups and the corresponding controls using Fisher's exact test (tests were conducted in R version 3.5.1, *R Core Team, 2019*).

### Functional roles of *dpr/DIP*s

Initially, several different combinations of one *dpr/DIP-Gal4* driver, and either a UAS-RNAi *dpr/DIP*, or a UAS-*dpr/DIP* expression transgene were assayed, using the intersectional genetic approach for visualization of small sets of *fru P1* neurons (*Figure 1C*; *Source data 5*). For the RNAi screen, parents laid eggs at 25°C for 2–3 days, and then the vials with eggs were transferred to 29°C, to increase effectiveness of RNAi constructs. For the over-expression screen, flies were raised at 25°C. Staining and confocal imaging was performed as described above. Through this initial screen, we found that knocking down *DIP-ε* in *DIP-α ∩ fru P1* neurons at 4–7 days was the only condition to yield a robust phenotype. Knockdowns were analyzed, with *DIP-ε* or *RFP* RNAi active in all *DIP-α* expressing cells. In addition, knockdowns were restricted to the visualized *fru P1 ∩ DIP-α* neurons with the use of *tub>GAL80>*.

The *fru P1 ∩ DIP-α* neurons were analyzed blind in 20 brains, in male and female controls and mutants, to determine the effect of *DIP-ε* knockdown on neuronal morphology (*Source data 5*). The presence or absence of morphological features were compared within sex between *DIP-ε* knockdowns and the corresponding control using a Fisher's exact test (R version 3.5.1, *R Core Team, 2019*).

## Acknowledgements

The work presented was supported by NIH grants awarded to MNA: R01GM073039, R01GM116998, R03NS090184. This work was also supported by funds from the Biomedical Sciences Department, College of Medicine, Florida State University. We are grateful for the support. We appreciate that colleagues sent *Drosophila* stocks (**Key Resources Table**). Stocks were also obtained from the Bloomington *Drosophila* Stock Center (NIH P40OD018537). Several antibodies used in this study were obtained from the Developmental Studies Hybridoma Bank, created by the NICHD of the NIH and maintained at The University of Iowa, Department of Biology, Iowa City, IA 52242. We thank Batory foods (Lithia Springs, GA) for the cornmeal used in the fly media. We appreciate experimental assistance from Catherina Artikis.

# Additional information

### Funding

| Funder | Grant reference number | Author |
| --- | --- | --- |
| National Institutes of Health | R01GM073039 | Savannah G Brovero |

Julia C Fortier
Hongru Hu
Pamela C Lovejoy
Nicole R Newell
Colleen M Palmateer
Ruei-Ying Tzeng
Michelle N Arbeitman

| National Institutes of Health | R03NS090184 | Ruei-Ying Tzeng |
| Florida State University | Department of Biomedical Sciences | Hongru Hu<br>Nicole R Newell<br>Colleen M Palmateer |
| National Institutes of Health | R01GM116998 | Savannah G Brovero<br>Hongru Hu<br>Pamela C Lovejoy<br>Michelle N Arbeitman |

The funders had no role in study design, data collection and interpretation, or the decision to submit the work for publication.

## Author contributions

Savannah G Brovero, Julia C Fortier, Hongru Hu, Pamela C Lovejoy, Nicole R Newell, Formal analysis, Validation, Investigation, Writing - original draft, Writing - review and editing; Colleen M Palmateer, Ruei-Ying Tzeng, Formal analysis, Validation, Methodology, Writing - original draft, Writing - review and editing; Pei-Tseng Lee, Resources; Kai Zinn, Resources, Writing - review and editing; Michelle N Arbeitman, Conceptualization, Formal analysis, Supervision, Funding acquisition, Writing - original draft, Project administration, Writing - review and editing

## Author ORCIDs

Kai Zinn  http://orcid.org/0000-0002-6706-5605
Michelle N Arbeitman  https://orcid.org/0000-0002-2437-4352

## Decision letter and Author response

Decision letter https://doi.org/10.7554/eLife.63101.sa1
Author response https://doi.org/10.7554/eLife.63101.sa2

## Additional files

### Supplementary files

• Source data 1. Image quantification. Data tables of quantification of *fru P1 ∩ dpr/DIP* neurons. This is an excel data table with 10 sheets that contain: 1. Raw Data Brain: raw data and observation notes only for areas of the brain scored/counted. 2. Raw Data VNC: raw data and observation notes for areas of the VNC scored/ counted. 3. Raw Scoring Data: raw scoring observations of areas in both brain and VNC scored. 4. Scoring Data Processed: assigning numerical values between 0 and 1 to scoring observations. 5. Raw Count Data: raw cell counts for areas in both the brain and VNC. 6. Cell Count Calculations: math done to cell counts so they were between 0 and 1. 7. Full Data Set: both male and female image analysis data with scoring observations and cell counts as values between 0 and 1. 8. Male Only: image analysis results for males (used to create heatmaps). 9. Female only: image analysis results for females (used to create heatmap). 10. Heatmap_Behavior_Plot: supplemental figure of the heatmap and behavior plot where the count data and scoring data has been separated. This sheet also contains additional heat maps of the male and female image count data.

• Source data 2. Behavioral data. Data tables of quantification of behavioral phenotypes. This is an excel data table with five sheets that contain all the behavioral data as follows: 1. READ ME: This first sheet is to explain the following four tabs. Please note that TNTQA1, as opposed to TNTQ is used in the following sheets. They both refer to the inactive form of TNT, but TNTQA1 is our lab nomenclature. 2. TrpA1 male-female courtship behavior data. 3. TrpA1 male alone courtship behavior data. 4. TNT courtship behavior data. 5. TNT DAM activity locomoter behavior data: This tab contains line crossing reads from *Drosophila* Activity Monitors (DAM).

• Source data 3. Single-cell RNA-seq. Data tables that include information from the single-cell RNA-seq study, including data matrices and UPSET analyses, in five sheets as follows: 1. Barcode rank plot and sequencing matrices. 2. Male 48 hr APF *fru P1* neurons dpr/DIP gene-cell barcode matrix of log normalized and scaled gene counts. Columns are cell barcodes, rows are *dpr/DIP* genes. 3. Male 48 hr APF *fru P1* neurons *dpr/DIP* expression matrix binarized by 0 or one where normalized, scaled expression >1 = 1 and expression <1 = 0. Data table is presented in transposed in format. Columns are *dpr/DIP* genes, rows are cell barcodes. 4. Numbers of single cells with increasing numbers of dprs/DIPs co-expressed, derived from data on sheet 3. 5. Upset plot of *dpr/DIP* expression combinations based on binarized expression matrix (sheet 3) where normalized, scaled expression >1 = 1 and expression <1 = 0. *dpr/DIP* expression dendrogram based on expression matrix presented in sheet 2.

• Source data 4. Sex hierarchy perturbation. Data tables of quantification of confocal data from sex hierarchy perturbations in *fru P1* ∩ *DIP-α* neurons. This excel file has the quantification from the sex hierarchy perturbation analyses, with the following sheets: 1. Read Me: This first sheet is to explain the following tabs and to display the features scored. Representative confocal 3D projections are shown below. 2. Genotype and Condition: Description of genotypes, conditions, expected perturbations and changes. 3A. Scoring Summary of set one overexpressors: Overexpression of sex hierarchy related genes in ALL DIP-α cells plus a loss-of-function line (Control: *DIP-α-GAL4; UAS>stop>GFP.Myr/+; fruFLP/+*). 3B. The corresponding scoring raw data of 4A. 4A. Scoring Summary of set two overexpressors (with tubGal80): Overexpression of sex hierarchy related genes in *DIP-α* and *fru P1* intersecting cells. (Control: *DIP-α-GAL4; UAS>stop>GFP.Myr/+; fruFLP/tub>GAL80>*). 4B. The corresponding scoring raw data of 5A (with *tubGal80*). 5. The conclusive tables of all scoring data and the corresponding statistics (p values from Fisher's Exact test; when p ≦ 0.05, the cell will be highlighted). 6. Confocal images showing staining controls: *UAS-Fru^M* constructs and expression pattern of *Dip-alpha* with *UAS-GFP*.

• Source data 5. RNAi and overexpression screen. Data tables of quantification of confocal data of RNAi and *dpr/DIP* overexpression analyses, in four sheets as follows: 1. READ ME: This first sheet is to explain the following four tabs. 2. *DIP-α-* and *DIPδ-GAL4* screen: This sheet contains all genotypes tested in the initial RNAi/overexpressor screen using both *DIP-α* and *DIP-δ* as the GAL4 driver. The controls are bolded. N = 5. This sheet contains the following columns: Genotype: Contains an abbreviated genotype, including the GAL4 driver and *dpr/DIP* knocked-down (RNAi) or overexpressed (UAS). Although not explicitly stated, all genotypes also contained a *UAS > stop > GFP.Myr* transgene as well as a *fru^FLP* transgene. Temperature Raised: The RNAi crosses were set up at 25°C where the flies were allowed to lay eggs for 2–3 days. Then, the parents were turned into a new vial, and the eggs were raised at 29°C. This was done to ensure that the RNAi was fully functional, as it was found to be less effective at 25°C. All overexpressor flies were raised at 25°C, as noted. Difficulty getting males: This was only an issue with the RNAi crosses at 29°C. Some of these crosses did not produce males at all, or produced far fewer males than females. Of note, these crosses always produced females with ease. Assayed at 16–24 hr: 'Yes' in this column indicated that five replicates per sex were dissected and stained as 16–24 hr adults. Assayed at 4–7 days: 'Yes' in this column indicated that five replicates per sex were dissected and stained as 4- to 7-day adults. Phenotype: Any phenotype observed when comparing the RNAi with its respective control. Notes: Relevant notes. 3. *DIP-α-GAL4; DIP-ε-RNAi*: Of all the genotypes tested, this was the only one to show a robust phenotype. The czi files were scored blind for conditions with and without *tub>GAL80>*. n = 20. A table summarizing the statistics is to the right of the raw data. See below for a more detailed description of how these phenotypes were scored. 4. *UAS DIP-α* screen: Although not explicitly stated, all genotypes also contained a *UAS > stop > GFP.Myr* transgene as well as a *fru^FLP* transgene. Each of the 7 GAL4s of interest were first crossed to *UAS DIP-alpha* (experimental) and Canton S (*white*) (control) with n = 5. The GAL4 stock was crossed to Canton S (*white*) (w;cs) as a control to eliminate balancers. The controls are bolded in this data sheet. 3 GAL4 drivers that showed potential phenotypes upon overexpressing *DIP-alpha* were identified for further study (*DIP-β*, *DIP-δ*, and *DIP-ε*). These genotypes were retested with an n = 15 and with *tub>GAL80>*. Upon further analysis, no robust phenotypes were observed and as a result, the images were not scored.

• Transparent reporting form

## Data availability

All raw data are provided in the supplementary materials. The sequencing data have been deposited in GEO under accession number GSE162098.

The following dataset was generated:

| Author(s) | Year | Dataset title | Dataset URL | Database and Identifier |
|---|---|---|---|---|
| Palmateer CM, Arbeitman MN | 2020 | Investigation of Drosophila fruitless neurons that express Dpr/DIP cell adhesion molecules | https://www.ncbi.nlm. nih.gov/geo/query/acc. cgi?acc=GSE162098 | NCBI Gene Expression Omnibus, GSE162098 |

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
