## [Decision Letter]

**Acceptance summary:**

*Drosophila* reproductive behaviors are directed by a subset of neurons that express the gene *fruitless*. This paper uses a diverse set of analyses and experiments to investigate the sexually dimorphic morphology and activity of a subset of these neurons characterized by expression of a family of cell:cell adhesion molecules, with the overarching suggestion that these adhesion molecules play a role in shaping the neuroanatomical differences that drive subtle behavioral differences.

**Decision letter after peer review:**

Thank you for submitting your article "Neurogenetic and genomic approaches reveal roles for *Dpr/DIP* cell adhesion molecules in *Drosophila* reproductive behavior" for consideration by *eLife*. Your article has been reviewed by two peer reviewers, and the evaluation has been overseen by Michael Eisen as the Senior Editor and Reviewing Editor. The reviewers have opted to remain anonymous.

The reviewers have discussed the reviews with one another and the Reviewing Editor has drafted this decision to help you prepare a revised submission.

This study by Brovero et al. identifies *DIP* and *Dpr* expression in *fruitless* positive neurons of the courtship circuits in the nervous system. *Fruitless* positive neurons/circuit shows sexual dimorphism in the number and morphology of the neurons, which is regulated by the function of male splice isoform of *fruitless* transcripts from the P1 promoter. One hypothesis is that sex-specific differences in the cell surface receptors might regulate morphological aspects of sexually dimorphic neuroanatomy within *fruitless* positive courtship circuit driving behavioral differences between sexes.

To test this, the authors use intersectional genetic approaches to determine sexually dimorphic *fruitless* positive neuronal subpopulations based on their combinatorial *DIP/Dpr* expression. They find that each *fruitless* positive neuron has a unique *DIP/Dpr* expression pattern. The expression patterns for some are sexually dimorphic. Elimination of sex specific splicing of *fruitless* or knock down of *fruitless^M^* disrupts sexual dimorphism in neuronal morphology. Genetically silencing or activating small subpopulations of *fruitless* positive neurons using the *DIP-/Dpr-GAL4fru^FLP^* intersectional methods, the authors also systematically delineate different neuronal contributions to different components of the courtship ritual. In addition, knock down of some DIPs also disrupt the sexually dimorphic neuroanatomy.

This paper systematically uncovers the expression of the large family of *DIP/Dpr* cell adhesion molecules in defining neuronal subpopulations within the courtship circuit, as well as their contribution to neuroanatomical differences of these neurons between males and females and their contribution to courtship behaviors.

While the paper reports a lot of solid data and analyses, there was a consensus among the editor and reviewers that the manuscript is unfocused and covers many things without going into depth in any, and as a result, the conclusions are not at a sufficiently high resolution to reveal much new about either the Fru circuit or *DIP/Dpr* function.

Essential revisions:

As described below, we would above all like to see a clearer and more focused revised manuscript, with the addition of some additional data to provide depth where the authors choose to pursue it. In particular, the revised manuscript should focus on the novel points you are trying to make and/or hypotheses they are trying to test, which were not clear in this submission. There is also interest in more clarity on the mechanism underlying the expression pattern of *DIP/Dpr* family members in *fruitless* positive neurons, in particular an answer to the question of whether fru regulates the expression of even a subset of *DIPs* and *Dprs* in different fru+ neurons in the circuit.

1) The experiments nicely show the sexual dimorphisms of *DIP/Dpr* expression profiles in subpopulations of fru positive neurons as well as the structure of these neurons. It is also nicely demonstrated that disrupting sex-specific splicing of *fru* as well as fru mutants disrupt the sexual dimorphism of these neuronal subpopulations. So it is clear that both *DIP/Dprs* and the morphology of the neurons are sexually dimorphic and likely are regulated by the sex determination pathway.

One question that arises from these experiments is whether *fru^M^* regulates the expression of *DIP/Dprs*. Can the authors pick a few of the *DIP/Dprs* and ask if their expression patterns are feminized in *fru^M^* mutants?

2) Subsection “Live tissue staining shows sexual dimorphism in the number of cells that overlap with *Dpr/DIP* binding and *fru P1* neurons”: What was the purpose of this experiment? Why didn't you use the existing antibodies and GAL4 T2A or MIMIC GFP lines?

3) Subsection “Activation of *fru P1* ∩ *dpr/DIP* neurons results in atypical courtship behaviors”: You are basically inactivating subsets of fru neurons and assaying their function in reproductive behaviors. this region is not relevant to *dip/dpr* function in those fru positive neurons, right? Maybe this needs to be spelled out better to set the context. The paper reads with the expectation that this experiment will be done to test *dip dpr* function in structure and function of the courtship circuits.

4) Subsection “Silencing *fru P1* ∩ *dpr/DIP* neurons result in courtship changes”: What is the difference between motor defects and motor impairment? Aren't they related?

5) "Not all the intersection genotypes with expression in the abdominal ganglion show the ejaculation phenotype" Are they labeling different subpopulations neurons in the abdominal ganglion or is this phenotypic difference due to GAL4 efficiency in driving TrpA1?

6) "Therefore, there are sex-differences in the co-expression patterns…" How do you explain this? Please elaborate.

7) Subsection “Changing the sex of *DIP-α* neurons alters the *fru P1* ∩ *DIP-α* co-expressing patterns”: Is it possible that maybe *fru^MC^* overexpression titrates out interactors or compete for binding sites for regulating differential sets of genes? Does fru dimerize and regulate itself? Can you elaborate a little more on why you get no phenotype in loss of function of *fru^MC^* compared to overexpression?

8) Figure 1A – the sex determination pathway is missing arrows in the PDF.

9) I understand that the authors analyzed different profiling studies for *DIP/Dprs*. But what accounts for the differences in the *DIP/Dpr* gene expression profiles in Figure 1B coming from different profiling studies? It would be good to elaborate either in the Results section or in the Discussion.

10) Structurally, I felt like the scRNAseq section perhaps belongs to the earlier part of the Results when the authors are describing expression patterns of *DIP/Dprs*.

11) What is the value of the data in Supplementary Figure 1 – the images are very hard to interpret and don't always make sense (e.g. why are the control Fru-labeled neurons not visible in the 48 hr pupa images?).

12) There were issues with file formats in several of the supplementary tables, e.g. the ones referenced as having the Upset plots (Source data 3), which we had trouble opening.

13) Will all the scRNA data be uploaded somewhere? How do these data compare to other single cell data of fly neurons (e.g. in the VNC, 10.7554/*eLife*.54074)?

14) Some of the choices of intersections for further study seem a bit arbitrary since several combinations that were not pursued in Figures 2 and 3 seem to also have less dense patterns. Is it possible to come up with a more principled set of criteria and numbers of labeled neurons, e.g. by using a nuclear marker?

---

## [Author Response]

Essential revisions:As described below, we would above all like to see a clearer and more focused revised manuscript, with the addition of some additional data to provide depth where the authors choose to pursue it. In particular, the revised manuscript should focus on the novel points you are trying to make and/or hypotheses they are trying to test, which were not clear in this submission. There is also interest in more clarity on the mechanism underlying the expression pattern of DIP/Dpr family members in fruitless positive neurons, in particular an answer to the question of whether fru regulates the expression of even a subset of DIPs and Dprs in different fru+ neurons in the circuit.

We thank the reviewers for their thoughtful comments that have improved our manuscript. Overall, the manuscript has been shortened and revised to focus on novel points and to clarify the hypotheses. One large editorial change is that we moved the single-cell RNA-seq data as requested.

1) The experiments nicely show the sexual dimorphisms of DIP/Dpr expression profiles in subpopulations of fru positive neurons as well as the structure of these neurons. It is also nicely demonstrated that disrupting sex-specific splicing of fru as well as fru mutants disrupt the sexual dimorphism of these neuronal subpopulations. So it is clear that both DIP/Dprs and the morphology of the neurons are sexually dimorphic and likely are regulated by the sex determination pathway.One question that arises from these experiments is whether fru^M^ regulates the expression of DIP/Dprs. Can the authors pick a few of the DIP/Dprs and ask if their expression patterns are feminized in fru^M^ mutants?

We did not perform the additional loss-of-function confocal microscopy experiments requested to determine if *fruP1* regulates *dpr/DIP* gene expression because we did not think these experiments would address the question. If the GFP pattern in the *fru P1* ∩ *dpr/DIP* neurons changed/feminized in the *fruP1* mutant background it does not prove that this is due to direct regulation of *dpr/DIP* gene expression by Fru^M^, just that Fru^M^ regulates something in those neurons that is required for wild type male morphology.

Additionally, we think that the five genome-wide studies that are summarized in Figure 1B address the question better, since they directly examine *dpr/DIP* gene expression in different *fruP1* perturbation conditions, and also Fru^M^ genomic binding sites. The genomics experiments in Figure 1B use both *fru P1* loss-of-function and *fru P1* gain-of-function to directly examine Fru^M^-regulated gene expression of all *dprs/DIPs*. One study examines *fru P1* cell-type specific gene expression in males and females (ribosome affinity profiling). In addition, two studies examine direct DNA binding of Fru^M^. An additional Fru^M^ computational binding site analysis is also presented in the supplementary materials. Given that the majority of the *dpr/DIPs* are significantly identified in at least three of these independent genomics studies provides strong and more comprehensive support that Fru^M^ regulates the expression of *dprs/DIPs* than microscopy experiments on a few genotypes. A more thorough description of these genomic-scale experiments is also now provided in the Results and Discussion.

We also do present an extensive and comprehensive analysis of sex hierarchy perturbations, including *fru P1* perturbations on one set of intersecting neurons. We present loss-of-function and gain-of-function studies, in both cell-autonomous and non-autonomous conditions, in *fru P1∩ DIP-α* neurons and show that these perturbations impact expression patterns, with feminization occurring (please see Figure 9). As prompted by a reviewer comment below, we also elaborate on the potential mechanisms (see comment #7). We also note that the field has comprehensively examined whether removing *fru P1* function from *fru P1* neurons results in feminized morphology. Two studies we cite throughout the paper were the first to systematically address this question and show feminization (Cachero et al., 2010; Yu et al., 2010)(Results). The neurons examined in these previous studies are in all the regions where the *fru P1* ∩ *dpr/DIP* GFP patterns are detected. We now make clear that these two previous studies systematically examined *fru P1* feminization phenotypes in *fru P1* mutants.

2) Subsection “Live tissue staining shows sexual dimorphism in the number of cells that overlap with Dpr/DIP binding and fru P1 neurons”: What was the purpose of this experiment? Why didn't you use the existing antibodies and GAL4 T2A or MIMIC GFP lines?

The live staining experiments were initiated before the GAL4 T2A/MIMIC *Drosophila* lines were available. There are not antibodies for each *Dpr/DIP*, as far as we know. The goal was to comprehensively assess *Dpr/DIP* binding to *fru P1*-expressing neurons, to gain a more systems-level understanding of the *Dpr/DIP* spatial distributions in the nervous system that is not afforded by the genomic analyses. Given the request to focus the paper, the live staining analysis is not in the current version of the paper, as it is not central to the study. These data are now referenced (Brovero et al., 2020) and so are available to the public.

3) Subsection “Activation of fru P1 ∩ dpr/DIP neurons results in atypical courtship behaviors”: You are basically inactivating subsets of fru neurons and assaying their function in reproductive behaviors. this region is not relevant to dip/dpr function in those fru positive neurons, right? Maybe this needs to be spelled out better to set the context. The paper reads with the expectation that this experiment will be done to test dip dpr function in structure and function of the courtship circuits.

We agree with the reviewer and made changes throughout the manuscript to have the expectations more concordant with the data presented, including a new title, and edits to all the sections.

4) Subsection “Silencing fru P1 ∩ dpr/DIP neurons result in courtship changes”: What is the difference between motor defects and motor impairment? Aren't they related?

They both are describing the same phenotype. We have now modified the manuscript so only “motor defect” is used to describe the phenotype where the male fly falls and is not able to quickly right himself in the courtship chambers. We also made it clear when we used the Trikinetics *Drosophila* Activity Monitor (DAM) assay to assess “locomotor activity”.

5) "Not all the intersection genotypes with expression in the abdominal ganglion show the ejaculation phenotype" Are they labeling different subpopulations neurons in the abdominal ganglion or is this phenotypic difference due to GAL4 efficiency in driving TrpA1?

Of the eight intersecting genotypes that have expression in the lateral protocerebral complex and mushroom body, the four that have the male ejaculation phenotype also have the highest number of neurons in the abdominal ganglion. This is now indicated in the Results. Given the tools in hand it is not possible to determine with certainty if these are the same subpopulations of neurons, as we would need to generate LexA driver tools that would be beyond the scope of this study.

There are intersecting genotypes with high numbers of neurons in the abdominal ganglion, but that do not show the ejaculation phenotype, as indicated in the Results. These genotypes do not have expression in the lateral protocerebral complex and mushroom body, which is one of the points we are making in this section—that the combinations of neurons are also critical.

While it is formally possible that differences in TrpA1 expression levels could be contributing to differences in behavioral outcomes, our tests of expression indicated that TrpA1 levels (detected by myc antibody staining) followed the levels/patterns of GFP expression in a large set of genotypes. We could not find reports in the literature that describe whether there is threshold amount of TrpA1 expression needed to generate neuronal activation. We note this caveat to our interpretation (Results).

6) "Therefore, there are sex-differences in the co-expression patterns…" How do you explain this? Please elaborate.

To focus the paper the section on correlation analyses of the expression patterns has been shortened. We no longer make the statement noted above.

7) Subsection “Changing the sex of DIP-α neurons alters the fru P1 ∩ DIP-α co-expressing patterns”: Is it possible that maybe fru^MC^ overexpression titrates out interactors or compete for binding sites for regulating differential sets of genes? Does fru dimerize and regulate itself? Can you elaborate a little more on why you get no phenotype in loss of function of fru^MC^ compared to overexpression?

We had described *fru^MC^* loss-of-function as having a less severe phenotype than Fru^MC^ overexpression due to the fact that the overexpression of Fru^MC^ is in the broad *DIP-alpha-Gal4* pattern, whereas the loss-of-function is only in *fru P1* neurons. The reviewer raises good points about the possibility that overexpression may also have other impacts. We now note these additional ideas the reviewer suggested in the paper (Results).

To address the review question: Fru^M^ is a member of the BTB-zinc finger family. The BTB domain has predicted dimerization functions, so it has been postulated that Fru^M^/Fru isoforms function as a dimer (either homodimer or heterodimer). To our knowledge, Fru^M^/Fru dimerization has not been demonstrated in published manuscripts.

8) Figure 1A – the sex determination pathway is missing arrows in the PDF.

Thank you for pointing this out. We will be sure that the manuscript conversion to pdf includes the arrow on publication.

9) I understand that the authors analyzed different profiling studies for DIP/Dprs. But what accounts for the differences in the DIP/Dpr gene expression profiles in Figure 1B coming from different profiling studies? It would be good to elaborate either in the Results section or in the Discussion.

Thank you for this suggestion. We have now added text to indicate the different types of genomic studies: Gene expression studies were *fru P1* loss-of-function, Fru^M^ gain-of-function (overexpression), and *fru P1* cell-type specific expression in males and females. The additional two genomic studies assess direct Fru^M^ binding targets. These studies were also done in different tissues/time points and have different statistical criteria. We considered a gene to be significant if the authors from the original study called the gene significant by their criteria. We have added text to the Results and Discussion to make clear that given the many types of analyses, genomic tools and statistical criteria it is not unexpected that each study may have some differences.

10) Structurally, I felt like the scRNAseq section perhaps belongs to the earlier part of the Results when the authors are describing expression patterns of DIP/Dprs.

Thank you for this suggestion. The single cell section is now after the results describing the meta-analysis of the GFP expression patterns with the behavioral data. The analysis of *fru P1∩ DIP-α* GFP expression with sex hierarchy and *dpr/DIP* perturbations follows.

11) What is the value of the data in Supplementary Figure 1 – the images are very hard to interpret and don't always make sense (e.g. why are the control Fru-labeled neurons not visible in the 48 hr pupa images?).

Given the advice to focus the paper, this supplementary figure is no longer part of this version of the manuscript. It is published in Brovero et al., 2020 (see response to comment 2 please). The version the reviewer saw had z-stack projections where the 48 hr pupa control sample were imaged starting from the posterior of the brain. The experimental samples were imaged with the anterior of the brain as the first section of the presented z-stack projection. This did not impact the cell quantification presented in the graphs.

12) There were issues with file formats in several of the supplementary tables, e.g. the ones referenced as having the Upset plots (Source data 3), which we had trouble opening.

We apologize that the reviewer was not able to open this file. On resubmission we will check with the *eLife* team to make sure all the files can be opened.

13) Will all the scRNA data be uploaded somewhere? How do these data compare to other single cell data of fly neurons (e.g. in the VNC, 10.7554/eLife.54074)?

The scRNA-seq data has been uploaded to the GEO repository (GSE162098) and this is provided in the Materials and methods section. There are no other data sets that examine pupal neurons. The one the reviewer points to is from the adult, and there are additional adult studies. These studies find *dpr/DIP* as cluster marker genes, or genes that have enriched expression in a cluster of neurons. We have added this information to the Results and Discussion.

14) Some of the choices of intersections for further study seem a bit arbitrary since several combinations that were not pursued in Figures 2 and 3 seem to also have less dense patterns. Is it possible to come up with a more principled set of criteria and numbers of labeled neurons, e.g. by using a nuclear marker?

We have modified the text to provide additional rationales:

“After examining all the *fru P1∩ dpr/DIP* patterns, and the single-cell RNA-seq data, it became apparent that *fru P1∩ DIP-α* neurons are sexually dimorphic, and that this is one of the genotypes with the fewest cells among the genotypes that were scored (Source data 1). Additionally, the *fru P1∩ DIP-α* neurons have arborization patterns that facilitate analysis of sex-differences in fine-scale processes that would be obscured in intersecting genotypes with broad expression.”

“To determine the functional roles of *dprs/DIPs* in *fru P1*-expressing neurons, we conducted an RNAi and over-expressor screen. We use the *DIP-α* and *DIP-δ* drivers, given that they have the most restricted intersecting expression patterns, with the fewest neuronal cell bodies among the genotypes scored (Source data 1), which facilitates visually identifying altered patterns in *fru P1∩ DIP* neurons, as discussed above.”